Resource

# Pan-vaccine analysis reveals innate immune endotypes predictive of antibody responses to vaccination

Slim Fourati [1], Lewis E. Tomalin [2], Matthew P. Mulè[3,4], Daniel G. Chawla [5], Bram Gerritsen [5], Dmitry Rychkov[6], Evan Henrich[7], Helen E. R. Miller[7], Thomas Hagan [8], Joann Diray-Arce[9,10], Patrick Dunn [11], The Human Immunology Project Consortium (HIPC)*, Ofer Levy [9,10], Raphael Gottardo[7,12,13], Minnie M. Sarwal[6], John S. Tsang [3], Mayte Suárez-Fariñas[2], Bali Pulendran [8], Steven H. Kleinstein [5,22]✉ and Rafick-Pierre Sékaly [1,22]✉

Several studies have shown that the pre-vaccination immune state is associated with the antibody response to vaccination. However, the generalizability and mechanisms that underlie this association remain poorly defined. Here, we sought to identify a common pre-vaccination signature and mechanisms that could predict the immune response across 13 different vaccines. Analysis of blood transcriptional profiles across studies revealed three distinct pre-vaccination endotypes, characterized by the differential expression of genes associated with a pro-inflammatory response, cell proliferation, and metabolism alterations. Importantly, individuals whose pre-vaccination endotype was enriched in pro-inflammatory response genes known to be downstream of nuclear factor-kappa B showed significantly higher serum antibody responses 1 month after vaccination. This pro-inflammatory pre-vaccination endotype showed gene expression characteristic of the innate activation state triggered by Toll-like receptor ligands or adjuvants. These results demonstrate that wide variations in the transcriptional state of the immune system in humans can be a key determinant of responsiveness to vaccination.

Prophylactic vaccination is a cost-effective strategy to prevent or reduce the effect of viral and bacterial infections. Vaccine efficacy often varies in the population and can depend on age[1], sex[2], ethnicity[3] and genetics[4,5]. Human immune responses are also shaped by the environment, including previous pathogenic perturbation of the immune system. Indeed, pre-vaccination predictors of antibody response to specific vaccines such as influenza, yellow fever and hepatitis B vaccines have been identified[6–9], as well as pre-vaccination predictive signatures spanning both influenza and yellow fever vaccines[10]. However, whether

pre-vaccination markers exist for all vaccine platforms or if universal pre-vaccination markers of vaccine response can be identified have not been addressed for a large number of vaccines.

To define the biological signatures associated with the induction of protective immune responses induced by vaccination, high-throughput transcriptomic technologies (microarray and RNA sequencing) have been used to profile the peripheral blood cells of vaccine recipients. Paired with the use of machine-learning techniques, previous studies have identified signatures (that is, sets of genes) of vaccine-conferred

A full list of affiliations appears at the end of the paper. *A list of authors and their affiliations appears at the end of the paper.
✉e-mail: steven.kleinstein@yale.edu; rafick.sekaly@emory.edu

protection and/or of protective antibody responses to immunization. For example, different aspects of pre-vaccination states, including the frequency of B cell subsets as well as the expression of genes related to B cell receptor signaling and antigen processing predicted antibody response to influenza, yellow fever and hepatitis B vaccinations[6,9–11]. In contrast, pre-vaccination expression of genes related to granulocytes and interferon-stimulated genes (ISGs) have been associated with a poor response to hepatitis B vaccination[6,12]. Genes related to apoptosis and inflammatory responses were also shown to be expressed at a higher level by participants with a better response to the influenza vaccine[7,13] and worse response to the malaria vaccine[14]. However, a common pre-vaccination signature shared by all these vaccines has yet to be identified. Moreover, some of the biological pathways identified showed opposite associations with response between vaccines (for example, interferon signaling is a negative predictor of antibody response for hepatitis B[12] but type I interferon genes are positive predictor of antibody response for influenza and yellow fever vaccination[10]), or between studies for the same vaccine (for example, B cell signaling for influenza vaccination[11,13]). The interpretation of these differences can often be complicated by not only the vaccine type, but also factors such as geographical region (for example, whether the targeted pathogen is endemic versus not), age and different genes in the same pathway (or gene set) driving the association signals. The interaction of those various factors is complex, and their effect could thus elude robust detection using smaller-size cohort studies. Meta-analyses, leveraging information from multiple cohorts, can increase the statistical power to detect pre-vaccination signatures predictive of antibody responses to vaccines despite potentially confounding variables (for example, age, ethnicity and geographical region).

Identifying a universal pre-vaccination signature predictive of antibody responses to vaccines and understanding the biological pathways associated with, and therefore potentially required for, inducing a protective humoral response following vaccination in healthy adults may lead to more effective strategies (for example, administration of immunomodulators) to enhance vaccine response[15]. Those new strategies may particularly benefit the most vulnerable populations, including infants, older people and immunosuppressed individuals.

Here, we show that a common pre-vaccination peripheral blood transcriptional signature is predictive of antibody responses across 13 different vaccines. Functional annotation of this signature shows enrichment of effector genes of pro-inflammatory responses and pre-exposure sensing of ligands associated with bacterial infections. Analysis of existing single-cell transcriptomic data from healthy participants showed that nonclassical monocytes and myeloid dendritic cells (DCs) are the likely sources of this pre-vaccination signature. The overlap in genes between this predictive signature and the transcriptomic signature following Toll-like receptor (TLR) stimulation or adjuvant treatment suggests the existence of naturally occurring pre-vaccination innate immune activation states potentially overlapping with inflammatory activation induced by these immune stimulants, which are associated with better responses to vaccination.

## Results

### Heterogeneity of transcriptional profiles before vaccination

Transcriptomic profiles of whole blood and peripheral blood mononuclear cells (PBMCs) of 820 adults aged 18 to 55 years before and after vaccination were collected from publicly available databases (referred to as the 'Immune Signatures Data Resource'[16]). Several vaccine platforms ranging from live viruses (that is, yellow fever, smallpox and influenza vaccines), inactivated viruses (that is, influenza vaccine) and glycoconjugate vaccines (that is, pneumococcal and meningococcal vaccines) were included in this dataset (Fig. 1a,b and Supplementary Table 1). We assessed the contribution of different sociodemographic (age, biological sex and ethnicity) and experimental (vaccine, defined here as unique combinations of vaccine platform and the targeted pathogen;

time after vaccination) variables on the variance in the transcriptomic data (Fig. 1c). Age (14%), time points (9%) and vaccine (9%) explained only a small fraction of the variance observed in the transcriptomic data; over 62% of the variance between samples remained unexplained by any of the recorded clinical and experimental variables.

### Pre-vaccination endotypes of the immune system

To understand the source of the variance between participants, we restricted our analysis to the pre-vaccination time points (Extended Data Fig. 1). We next used hierarchical clustering to identify subgroups of participants with similar transcriptomic profiles before vaccination.

Hierarchical clustering (an unsupervised method) followed by identification of the optimal number of clusters by the Gap statistic identified three groups of participants (that is, endotypes) based on their pre-vaccination expression of gene sets included in the MSigDB hallmark gene sets[17] and blood transcriptomic modules (BTMs[18]; Fig. 2 and Extended Data Fig. 2a). Neither age (Kruskal–Wallis test, $P = 0.597$), sex (Fisher's exact test, $P = 0.570$), nor preexisting antibody levels to the immunogen (Kruskal–Wallis test, $P = 0.103$) were significantly associated with the differences in gene expression observed in these three endotypes (Extended Data Fig. 2b). Using samples collected 7 d before vaccination, those just before vaccination (day 0) and those collected at day 70 or beyond after vaccination from the same participants ($n = 74$), we calculated the temporal stability metric[10] and confirmed the relative stability of these transcriptomic profiles over time (Extended Data Fig. 2c; temporal stability metric = 0.73).

One endotype showed heightened expression of transcriptomic markers of monocytes and DCs, ISGs and pro-inflammatory genes and thus was designated a high inflammatory (inflam.hi) endotype. Transcriptomic markers of monocytes and DCs induced in the inflam.hi endotype included several genes encoding innate immune sensors (*TLR1*, *TLR2* and *TLR4*) and also genes of the TLR4 signaling cascade (*TLR4*, *LY96*, *DNM3* and *PLCG2*; Extended Data Fig. 2d). The type I interferon signaling cascade was also an important feature of the inflam.hi endotype. Receptors upstream of the interferon pathways (encoded by *IFNA2*, *IFNAR1*, *IFNAR2* and *TYK2*), nucleic acid sensors that trigger this pathway (encoded by *RIGI*, *TRIM25*, *MAVS*, *TRAF6* and *TANK*), and transcription factors that regulate the expression of ISGs (encoded by *STAT1*, *STAT2*, *IRF1* and *IRF7*) were all upregulated in the inflam.hi endotype compared to the other two endotypes. The nuclear factor-kappa B (NF-κB) pathway, whose activation is a hallmark of inflammation, and its target genes, including pro-inflammatory cytokines (*TNF*, *IL6* and *IL1B*) and their receptors (*TNFRSF1A*) or effector molecules regulated by NF-κB, including the metalloprotease ADAM17 that cleaves the ectodomain of tumor necrosis factor (TNF), were all elevated in the inflam.hi endotype. Likewise, the interleukin (IL)-6 signaling pathway (encoded by *IL6R*, *JAK2* and *STAT3*), a pathway that triggers the proliferation of activated B cells, was increased in the inflam.hi endotype. Moreover, several genes of the inflammasome complex and IL-1 signaling, also downstream of NF-κB, were also upregulated in these participants, including *IL1A*, *IL1B*, *IL1R1* and *IL1RAP*. Altogether, this endotype was characterized by genes and pathways involved in pro-inflammatory processes common to nucleic acid sensing, which could promote the development of an immune response to vaccines.

A second endotype showed significantly lower expression of the above-listed pro-inflammatory genes and pathways (that is, NF-κB and ISGs) when compared to the first endotype (Supplementary Table 2). This endotype was designated as the low inflammatory (inflam.lo) endotype. Heightened expression levels of of transcriptomic markers of natural killer (NK) cells, T cells, B cells and target genes of the transcription factors E2F and MYC both involved in the upregulation of cell proliferation and cell metabolism were features specific to the inflam.lo endotype. Transcriptomic markers of NK cells induced in the inflam.lo endotype included cell surface markers of NK cells (encoded by *KLRD1* and *KLRB1*), effector molecules of cytotoxic function (encoded

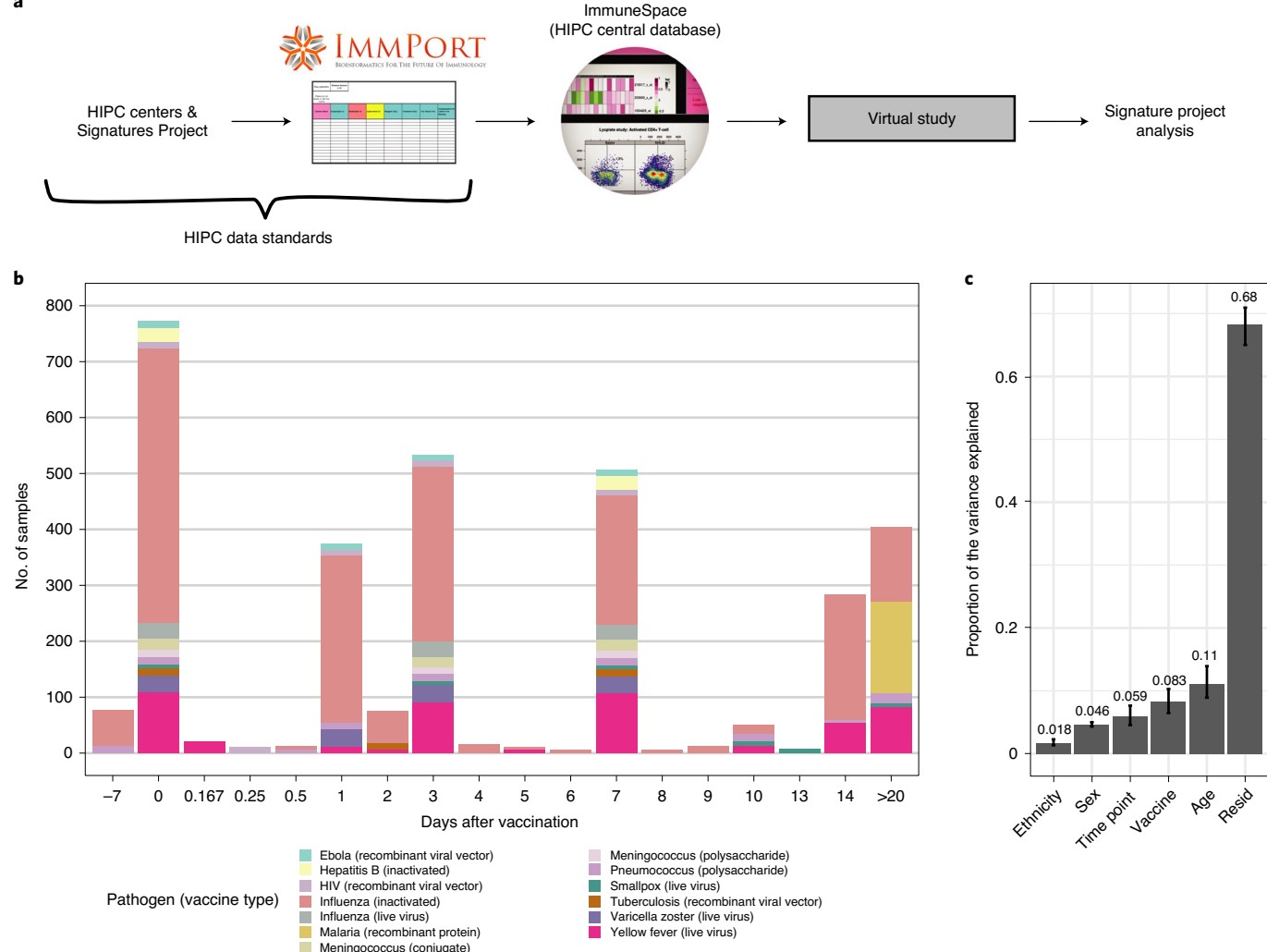

**Fig. 1 | Creation of a combined dataset of transcriptional responses to vaccination across diverse vaccine platforms and target pathogens.**
**a**, Flowchart describing the collection, curation, standardization and preprocessing steps leading to the creation of the vaccine transcriptomics compendium. **b**, Histogram of the time points before (days −7 and 0) and after (days > 0) vaccination available in the Immune Signatures Data Resource. In the plot, each vaccine is represented by a different color, while the size of the bar is proportional to the number of samples with available transcriptomic data.

Only adults aged 18–50 years, with available pre-vaccination data were included in the resource. **c**, Principal variance component analysis was used to estimate the proportion of the variance observed in the transcriptomic data that can be attributed to clinical (age, sex, ethnicity) and experimental variables (time after vaccination, vaccine). The proportion of the variance that could not be explained by those variables is depicted by the residuals (resid). Confidence intervals (95%, percentile method) and bar height (mean) were computed from 4,000 bootstrap replicates.

by *GZMB*, *FASLG* and *CASP3*), and genes of the IL-12 signaling cascade (*IL12RB1* and *STAT4*). Transcriptomic markers of T cells expressed in the inflam.lo endotype included members of the IL-2 signaling cascade (encoded by *IL2RA*, *IL2RB* and *LCK*), CD28-dependent PI3K–AKT signaling cascade (encoded by *CD28*, *CD80*, *PIK3CA*, *PIK3R1*, *PIK3R3* and *AKT3*) and IL-7 signaling cascade (encoded by *IL7* and *IL7R*); the latter two pathways being involved in the maintenance of the naïve T cell pool. Transcriptomic markers expressed by B cells of the inflam.lo endotype included cell surface receptors (encoded by *CD79A*, *CD79B*, *CD22* and *CD19*) and kinases (encoded by *FYN* and *BTK*) of the B cell receptor signaling complex. Known target genes of E2F and MYC induced in the inflam.lo endotype include cell cycle and proliferation regulators (*MYC*, *CDKN2A* and *AURKA*) and cell metabolism (*LDHA*, *MTHFD2* and *TYMS*). Altogether, this endotype was characterized by the lower expression of genes downstream of innate sensing (that is, interferons and NF-κB target genes).

Finally, a third endotype showed a mixed transcriptomic profile between inflam.lo and inflam.hi endotypes and was designated as

the middle inflammatory (inflam.mid) endotype. T cell-specific, NK cell-specific and B cell-specific genes were significantly upregulated in these participants compared to the inflam.hi endotype and significantly higher levels of pro-inflammatory genes were found in this endotype compared to the inflam.lo endotype (Supplementary Table 2).

**Immune cell frequencies vary between the endotypes**
Flow cytometry (*n* = 164) and immune cell deconvolution[19,20] were used to determine if the three pre-vaccination inflammatory endotypes were driven by the frequency of different innate and adaptive immune cell subsets (Extended Data Fig. 2e). The inflam.lo endotype showed an increased frequency of naive B cells (CD19+CD27−IgG−IgA− cells with heightened expression of ABCB4, ADAM28 and BACH2)[20], which is in line with the above-described gene expression profiles (Fig. 2). CD8+ T cells (CD3+CD8+CD45RA+ cells with heightened expression of CRTAM, PIK3IP1 and TRAV12-2) were also more prevalent in this endotype. In contrast, the inflam.hi endotype showed a statistically significant increase in monocyte frequencies (19% of immune

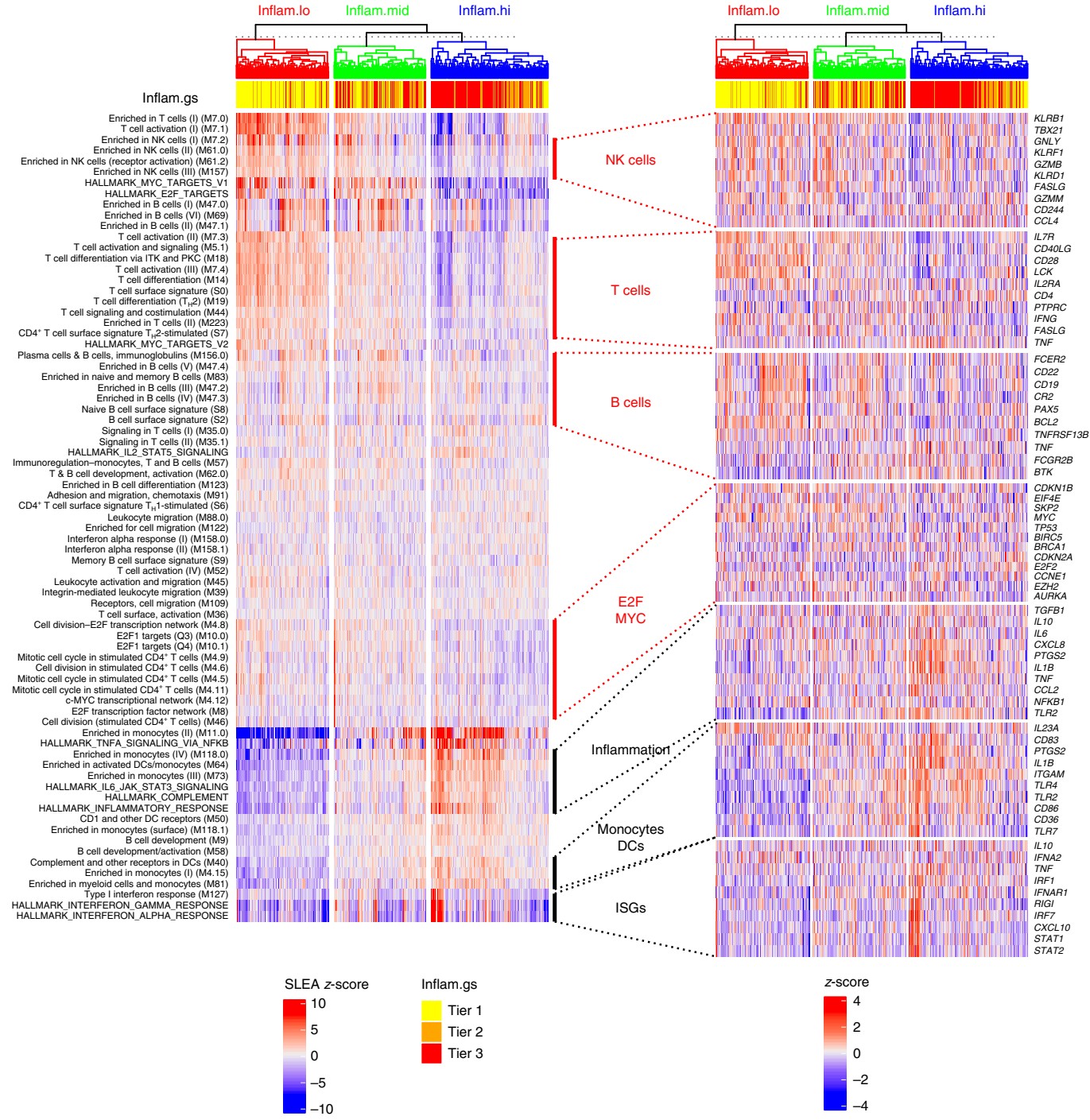

**Fig. 2 | Participants have distinct pre-vaccination transcriptomic profiles.** Hierarchical clustering (Euclidean distance metric and complete linkage agglomeration method) of pre-vaccination samples (day −7 and day 0) based on the expression of the BTMs and hallmark gene sets. The overall transcriptomic activity of gene sets/modules was estimated using sample-level enrichment analysis (SLEA). Three groups of participants/endotypes can be identified by cutting the dendrogram. Average SLEA score of the four hallmark inflammatory gene sets (bold row labels; inflam.gs), discretized in tertiles, is shown as sample annotation. Endotypes were designated as high (inflam.hi), low (inflam.lo) and middle (inflam.mid) inflammatory pathways. For each of the seven supersets of hallmark and BTM gene sets, ten canonical genes annotated to NK cells, T cells, B cells, E2F/MYC, inflammation, monocytes/DCs and ISGs, respectively (heat map). $T_H2$, type 2 helper T cell.

cells in inflam.hi versus 16% in inflam.lo; Wilcoxon rank-sum test, $P = 7.75 \times 10^{-5}$), in line with the results from the transcriptomic profiling (Fig. 2). To assess whether the change in gene expression between the three endotypes could be explained solely by the difference in immune cell frequency, differential expression analysis was performed, adjusting for the immune cell frequency, and reidentified inflammatory genes as markers of three endotypes (Supplementary Table 3). This analysis suggests that the difference in inflammatory gene expression between the three endotypes could not be explained by differences in cell frequencies alone and confirmed the differential transcriptomic activity of those inflammatory genes between endotypes.

## Endotypes modulate the transcriptional response to vaccines

Next, we evaluated the impact of the pre-vaccination inflammatory endotypes on the magnitude and kinetics of post-vaccination transcriptional responses. The pre-vaccination inflammatory endotypes explained 12.5% of the variance in gene expression observed before and after vaccination (Extended Data Fig. 3a), independently of age and sex of participants. On average, participants from the inflam.hi endotype, which had the highest pre-vaccination levels of pro-inflammatory pathways, showed reduced vaccine-induced expression of pro-inflammatory pathways (for example, complement pathway and IL-6 signaling pathway) at days 1 and 3 after vaccination when compared to the participants from the inflam.lo ($\log_2$ fold change ($\log_2$FC) < −1.46; Wilcoxon rank-sum test, $P$ < 0.0106) and inflam.mid ($\log_2$FC < −0.643; Wilcoxon rank-sum test, $P$ < 0.0996; Fig. 3a and Extended Data Fig. 3b) endotypes. By day 7, levels of the pro-inflammatory pathways returned to pre-vaccination levels in all three endotypes and levels remained as such over the duration of the follow-up. Similarly, participants from the inflam.hi endotype showed reduced induction of ISGs at day 1 after vaccination when compared to the inflam.lo ($\log_2$FC = −2.81; Wilcoxon rank-sum test, $P$ = $8.08 \times 10^{-4}$) and inflam.mid ($\log_2$FC = −1.54; Wilcoxon rank-sum test, $P$ = 0.0996; Fig. 3b and Extended Data Fig. 3c) endotypes. The participants from the inflam.hi endotype also had a lower B cell signature on day 7 and beyond compared to participants from the inflam.lo endotype ($\log_2$FC = −0.866; Wilcoxon rank-sum test, $P$ = $1.87 \times 10^{-4}$; Fig. 3c and Extended Data Fig. 3d). The levels of B cell markers returned to pre-vaccination levels by day 7 in the inflam.lo group contrary to the inflam.hi endotype where B cell markers were sustainably induced compared to pre-vaccination levels (Extended Data Fig. 3e). Similarly, type 2 helper T cell markers, necessary to mount a humoral response, were induced at day 7 after vaccination in the inflam.hi group but not in the inflam.lo (Fig. 3d and Extended Data Fig. 3f). The inflammatory endotypes affected the magnitude of the transcriptomic changes triggered by the vaccines, specifically at the earliest time points. However, peak responses occurred at the same time points in all three endotypes.

## Universal signatures predict vaccine antibody responses

We then assessed the association between the pre-vaccination endotypes and antibody responses triggered by each one of the 13 vaccines included in this study and measured approximately 1 month after immunization (by hemagglutination-inhibition, enzyme-linked immunosorbent or neutralizing assays; Supplementary Table 1). Participants from the inflam.hi endotype showed significantly higher antibody responses across all vaccines compared to participants of the inflam.lo endotype ($\log_2$FC = 0.253, Wilcoxon rank-sum test, $P$ = 0.00439; Fig. 4a) and inflam.mid endotype ($\log_2$FC = 0.167, Wilcoxon rank-sum test, $P$ = 0.0595). The association between the inflammatory endotypes and the antibody response was statistically significant for influenza inactivated vaccines and exhibited a similar trend for the other vaccines included in our study (Extended Data Fig. 4a). The inflammatory endotypes also tended to be associated with antibody response measured beyond day 28 but did not reach significance (Extended Data Fig. 4b). The association of inflammatory endotype with antibody response did not significantly differ between assays used to assess antibody response (hemagglutination-inhibition, enzyme-linked immunosorbent, or neutralizing assays; likelihood-ratio test, $P$ = 0.265). The inflammatory endotypes were not predictive of the magnitude of the humoral response to influenza, hepatitis B and varicella zoster vaccines in older people (aged 50 years and above; Extended Data Fig. 4c). Taken together, pre-vaccination immunological endotypes were associated with the magnitude of the vaccine-induced antibody response in adults.

To complement the unsupervised approach, we used a supervised approach to identify genes that are predictive of high (top 30%) versus low (bottom 30%) antibody response to vaccination. We trained a random forest classifier that predicts vaccine-specific antibody responses based on pre-vaccination gene expression profiles. This classifier achieved an area under the receiver operating characteristic (ROC) curve (auROC) of 62.3% as estimated by tenfold cross-validation (Fig. 4b). The accuracy of the classifier was greater for the vaccines with the largest number of samples (influenza inactivated, $n$ = 335, auROC = 63.0%; yellow fever, $n$ = 93, auROC = 51.6%) and deteriorated for vaccines with smaller sample sizes (Extended Data Fig. 4d; tuberculosis, $n$ = 8, auROC = 37.5%). The accuracy of the classifier was equal to vaccine-specific classifiers trained and tested on that same vaccine (Extended Data Fig. 4e). We did not observe any significant association between misclassification and the biological sex, age, ethnicities, geographical locations or assays used to measure antibody response of the participants, suggesting that the classifier accuracy is not affected by these parameters. For example, the yellow fever vaccine recipients included in the Immune Signatures Data Resource originated from five cohorts recruited in the United States, Canada, Switzerland, Uganda and China. The supervised classifier was significantly associated with high vaccine response in all cohorts except the one from the United States. The inflammatory signatures were predictive of antibody titers independently of the route of vaccination because our data sets include vaccines that were administered intramuscularly, intravenously or intranasally (for example, FluMIST).

The top 500 predictive genes selected by their importance in the classifier were enriched for inflammatory markers (50 genes of 500; Fisher's exact test, $P$ = $1.13 \times 10^{-11}$; Fig. 4c). Fourteen genes contributed to the majority (importance > 50%) of the classifier predictions (*KCNJ2*, *UTY*, *CNTNAP2*, *PTGS2*, *MAPK8IP1*, *LTC4S*, *ZNF124*, *EREG*, *CASP5*, *EGR1*, *CXCL10*, *ZNF248*, *DDX3Y* and *CCL20*). Those fourteen genes included pro-inflammatory cytokines and chemokines (*CXCL10*/IP-10, *CCL20*), mediators of IL-1, NF-κB signaling (*MAPK8IP1*, *CASP5* and *EGR1*)[21,22] and NF-κB target genes (*KCNJ2*, *PTGS2* and *ZNF248*)[23]. Those fourteen genes were compared to six previously identified pre-vaccination signatures of vaccine responses[6,7,10,12,24,25]. There was no significant overlap in gene content between those fourteen genes and the six previously identified pre-vaccination gene signatures (Extended Data Fig. 4f). Notably, the fourteen genes were the only ones to predict antibody response across most of the vaccines tested. In contrast, most of the previously identified signatures, including a pro-inflammatory signature we previously identified that predicted influenza vaccination response[7], were largely predictive for the vaccine types they have been trained on and less on the remaining vaccine types (Fig. 4d). Altogether, the signature identified here, heightened in the inflam.hi endotype, provides evidence that a specific inflammation signature pre-vaccination helps to mount a good antibody response across multiple vaccines.

## Cellular sources of the pre-vaccination endotypes

To identify the cells that potentially express the inflammatory genes included in the classifier of vaccine-induced antibody responses, we utilized publicly available CITE-seq (cellular indexing of transcriptomes and epitopes by sequencing) data from PBMCs collected from 20 healthy participants before vaccination with inactivated influenza vaccines[10]. We tested if the inflammatory genes were enriched in specific cell subsets or if their expression reflected a heightened global state of immune cell activation before vaccination common to multiple cell subsets. We analyzed the expression of the inflammatory genes of the classifier of vaccine-induced antibody responses within clusters of single cells defined by the expression of more than 80 specific cell surface proteins (Fig. 5a and Extended Data Fig. 5). The inflammatory genes intersecting between the unsupervised analysis (identified in Fig. 2) and supervised analysis were highly enriched within the innate immune cell subsets compared to other cell populations, specifically within the CD14$^+$CD16$^-$ classical monocytes and CD1c$^+$CD11c$^+$ myeloid DCs (Fig. 5b). These results highlight innate immune myeloid cells as the most likely cellular source of the pre-vaccination activated state found through both supervised and unsupervised analysis (as also suggested from Fig. 2).

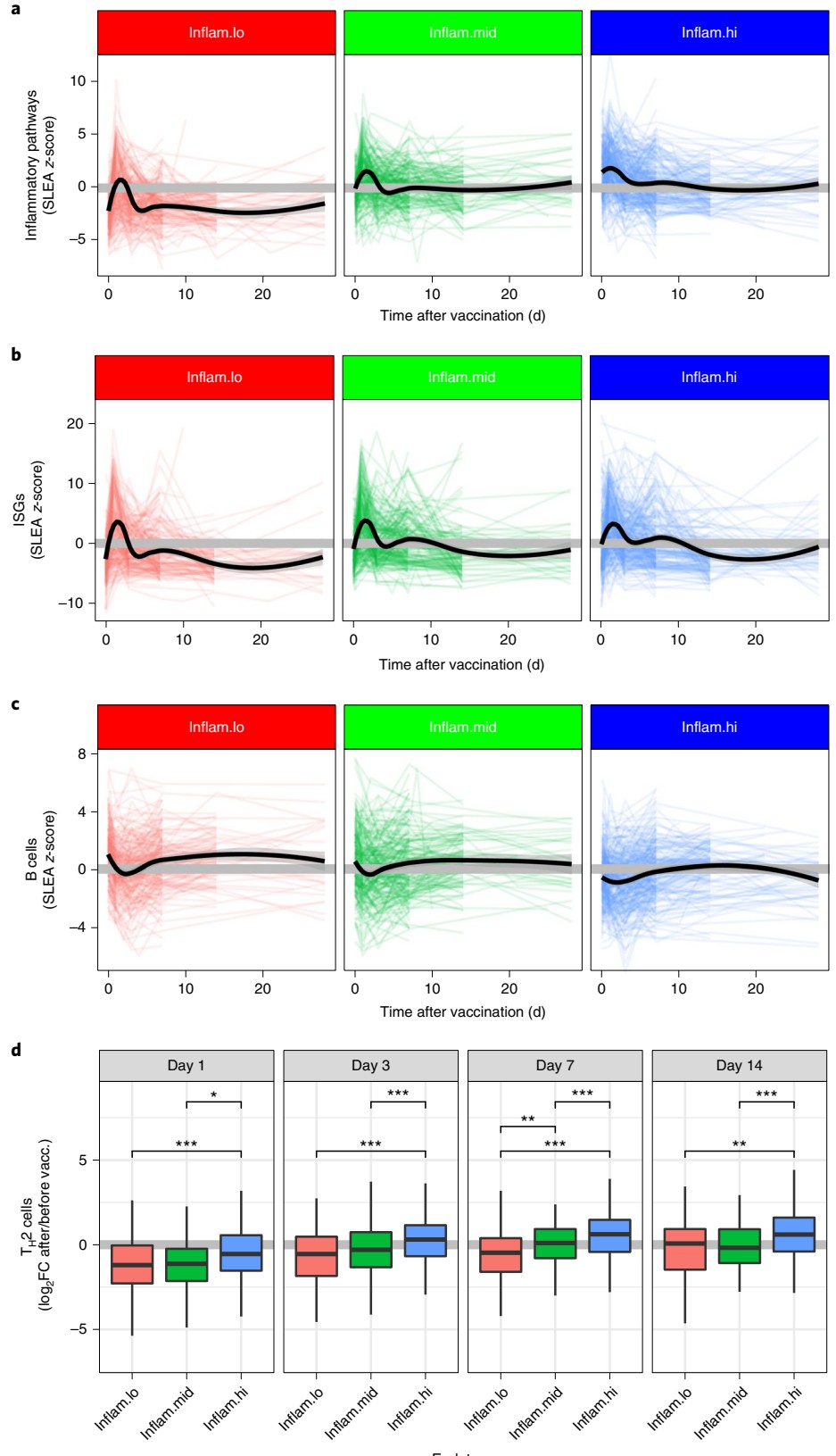

**Fig. 3 | Kinetics of the vaccine response are dictated by the pre-vaccination endotypes. a–c,** Line plots showing the expression of inflammatory pathways (**a**), ISGs (**b**) and B cells (**c**) as a function of time, separated by participants with low, middle or high pre-vaccination inflammation (inflam.lo, $n = 235$; inflam.mid, $n = 237$; inflam.hi, $n = 304$). Each colored line corresponds to one participant. LOESS regression was used to determine the average expression per pre-vaccination endotype (black lines). **d,** $T_H2$ cell markers fold change values over

pre-vaccination data for several time points after vaccination (day 1, inflam.lo $n = 117$, inflam.mid $n = 117$, inflam.hi $n = 139$; day 3, inflam.lo $n = 166$, inflam.mid $n = 165$, inflam.hi $n = 202$; day 7, inflam.lo $n = 159$, inflam.mid $n = 147$, inflam.hi $n = 198$; day 14 inflam.lo $n = 81$, inflam.mid $n = 103$, inflam.hi $n = 100$). For each box plot, the vertical line indicates the median, the box indicates the interquartile range, and the whiskers indicate 1.5 times the interquartile range. Wilcoxon rank-sum test; *$P < 0.05$, **$P < 0.01$ and ***$P < 0.001$.

Pre-vaccination inflammation in seemingly healthy participants can presumably arise from a noninfectious etiology or from potentially subclinical pro-inflammatory responses induced by bacteria or viruses. To identify the possible upstream signals associated with the inflammation described above, we used the seven-gene classifier described in work by Sweeney et al.[26] to discriminate between inflammatory signatures induced in response to pathogenic bacterial (classifier score above 0) or viral (classifier score below 0) infections. Applying this classifier to our cohort of vaccinees showed that participants within the inflam.hi endotype and the highest antibody-response group expressed genes that were more associated with exposure to bacterial infections (Fig. 6a).

We further observed that one of the bacterial markers in this seven-gene classifier, *TNIP1*, is a known NF-κB target and that the classifier score was positively correlated with an induction of NF-κB target genes. This contrasts with *IFI27*, an ISG used as a viral marker in the seven-gene classifier, and that interferon targets negatively correlated with the bacterial/viral classifier score. Interestingly, vaccines that were correctly predicted by the antibody-response classifier showed a stronger expression of NF-κB targets in high responders than low responders (Extended Data Fig. 6a; influenza inactivated, $\log_2 FC = 2.48$; yellow fever, $\log_2 FC = 0.743$; hepatitis B, $\log_2 FC = 1.12$). ISGs, downstream of interferon-regulatory factor 7 (IRF-7), were also associated with a robust humoral response to most of the vaccines except vaccines using poxvirus vectors such as the smallpox or yellow fever vaccines, for which strong expression of ISGs were associated with hyporesponse (Extended Data Fig. 6a). This analysis confirmed previous observations indicating that microbial elements (bacterial or viral) could be associated with the response to vaccines[27].

In support of these observations, we queried publicly available transcriptomic datasets related to bacterial inflammation[28], viral inflammation[28], pathogen recognition receptor (PRR) activation[29] and antibiotic treatment[27] to identify pathways associated with the prevalence of these inflammatory signatures that correlated with pan-vaccine antibody response. Again, counterintuitively, our inflammatory signature identified in seemingly healthy participants significantly overlapped with gene signatures from participants infected by *Staphylococcus aureus* and *Streptococcus pneumoniae* compared to healthy participants and to peripheral mononuclear cells stimulated in vitro with the TLR2/TLR6 ligand PAM2 (Extended Data Fig. 6b). Gene expression of DCs stimulated with bacterial (cGAMP, SeV, zymosan and lipopolysaccharide) and viral (polyIC) pattern-recognition ligands (several of them used as vaccine adjuvants)[30] showed strong induction of the inflammatory genes that were part of our classifier, suggesting that the heightened expression of those genes is a hallmark of a naturally adjuvanted immune system (Fig. 6b).

## Discussion

In this work, we characterize the interindividual heterogeneity in the inflammatory state of the peripheral immune system before vaccination and its association with vaccine response. Indeed, we identify three endotypes, inflam.hi, imflam.mid and imflam.lo, defined by multiple blood transcriptional signatures and a distinct distribution of cell subsets before vaccination. Our results show that these endotypes are associated with the relative magnitude of the antibody response across 13 different vaccines. Our work highlights the impact of the pre-vaccination immune system and suggests a role for pre-sensitization of the innate immune system to pathogen-associated molecular patterns in priming the B cell response to vaccination. The results presented here extend earlier definitions of pre-vaccination signatures to more diverse vaccines and populations; more importantly, they point to a framework that can lead to the inclusion of adjuvants that are more efficient at stimulating vaccine-induced protective immune responses.

Our approach consisted of training on all 13 vaccines and distinguishes this work from previously published reports. Importantly, the resulting classification model predicted the magnitude of the antibody response with a significant accuracy across these 13 vaccines. This strategy is likely the main factor that has contributed to the identification of this pan-vaccine classifier. Training on one vaccine type did not confer predictive power on distinct vaccine types irrespective of whether this was a live, inactivated or subunit vaccine. In contrast, the global classifier of vaccine responses identified herein performed as well as a classifier trained on any given vaccine and tested on that same vaccine. A similar finding is described in our companion paper by Hagan et al.[31] that focuses on post-vaccination response to the same 13 vaccines and identified a global transcriptomic signature associated with antibody response when all the vaccines are synchronized before building a classifier.

Our results show that qualitative and quantitative features, including transcriptional programs (MYC and E2F versus interferons and NF-κB target genes), can identify a pre-vaccination environment that leads to a heightened vaccine-induced antibody response. Expression of NF-κB, the prototypic transcription factor that controls the development of inflammatory responses, and its target genes are induced in the inflam.hi state. NF-κB is essential for driving the transcription of cytokines and chemokines (for example, *CXCL10*) that trigger cells of the innate and adaptive immune responses to migrate to sites of vaccination and differentiate into effector cells. Consistent with some previous reports on pre-vaccination signatures positively associated with antibody responses to vaccination[15], upregulation of ISGs is a feature of this state of participants, including IRF-7, the master transcription factor of the type I/type II interferons cascades. Type I and type II interferons regulate genes involved in antigen processing and presentation. The level of B cell responses in blood was

**Fig. 4 | Prediction of the antibody response by the pre-vaccination endotypes. a**, Box plot of the maximum fold change (MFC) antibody responses as a function of the pre-vaccination inflammation endotypes (inflam.lo, $n = 212$; inflam.mid, $n = 233$; inflam.hi, $n = 281$). The MFC was scaled to a mean of 0 and a standard-deviation of 1 across vaccines. For each boxplot, the vertical line indicates the median, the box indicates the interquartile range, and the whiskers indicate 1.5 times the interquartile range. A Wilcoxon rank-sum test without correction for multiple testing was used to assess differences in antibody response between the two endotypes; *$P < 0.05$, **$P < 0.01$ and ***$P < 0.001$. **b**, A supervised machine-learning approach was adopted to train a random forest classifier using pre-vaccination gene expression to distinguish high vaccine responders (top 70%) from low vaccine responders (bottom 30%). The predictive performance of the classifier was assessed by tenfold cross-validation (10-CV). The ROC curve is presented along with the auROC and 95% confidence intervals estimated from the tenfold CV. **c**, The top 500 predictive genes/features included in the classifier (importance > 0) overlapped with inflammatory genes identified in the unsupervised approach (two-sided Fisher's exact test, $P = 1.13 \times 10^{-11}$). Heat map showing the pre-vaccination expression of the overlapping genes. Samples (columns) are ordered by increasing expression level of the inflammatory genes. A Wilcoxon rank-sum test was used to assess the association between the inflammatory signatures and high/low antibody response and resulted in a $P$ value of 0.00265. **d**, Comparison of eight genes contributing the majority of the classifier prediction (importance > 50%) against previously identified pre-vaccination signatures of vaccine response. MetaIntegrator was used to calculate an auROC for each previously published pre-vaccination signature of vaccine response, as well as the eight genes identified in this work, using each of the transcriptomic studies within the Immune Signatures Data Resource. Circles correspond to studies that were used to train the pre-vaccination signatures, while asterisks indicate significantly better than random identification of high responders in each transcriptomic study as determined by a permutation test.

lower in the inflam.hi compared to inflam.lo group, suggesting that antibody-producing B cells migrate to tissues instead of remaining in circulation. In contrast, inflam.lo participants demonstrated the

upregulation of transcriptional networks that highlight genes and pathways of T cell and B cell activation and proliferation including a heightened expression of the E2F and MYC transcriptional programs

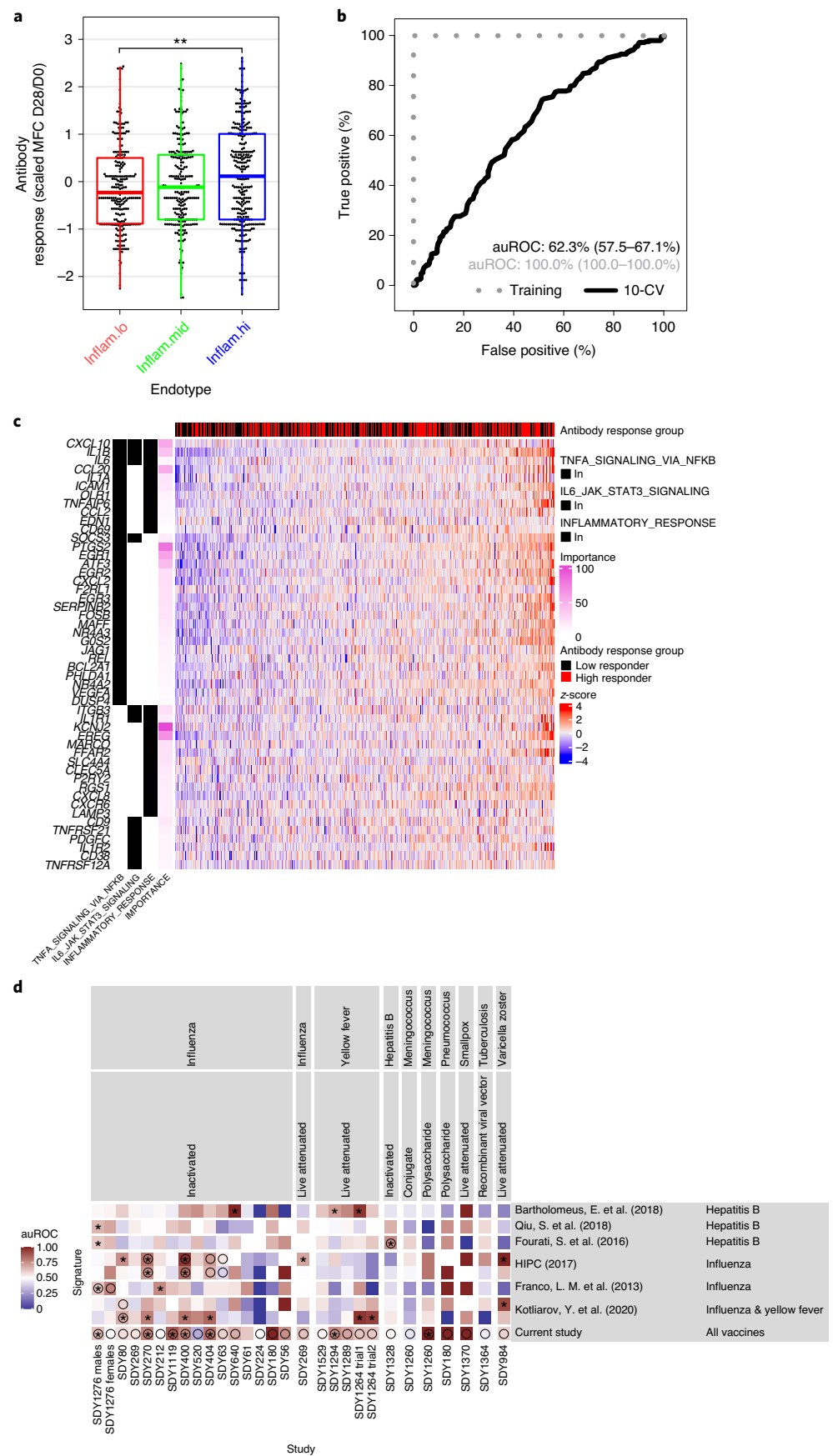

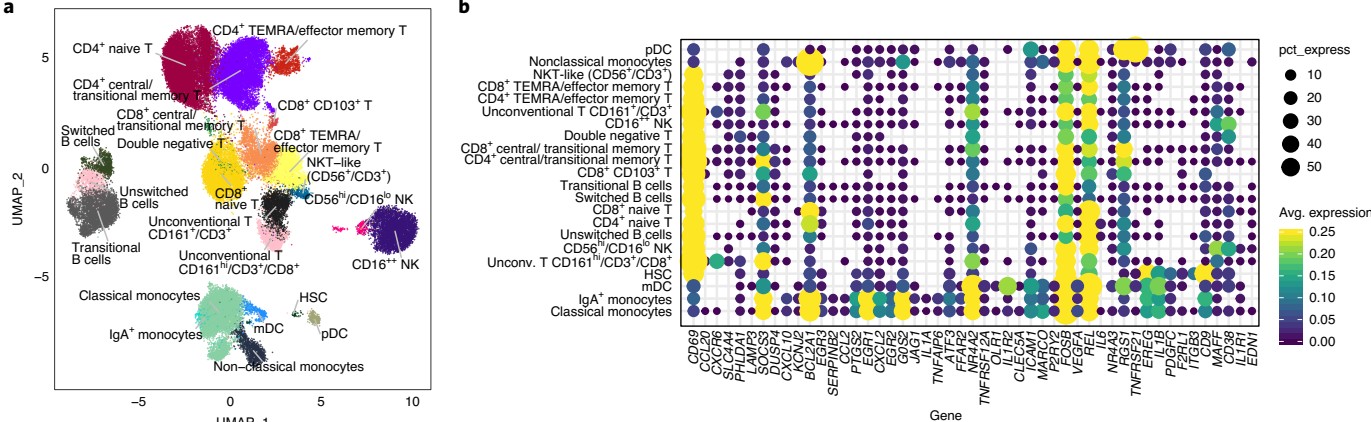

**Fig. 5 | a, Pre-vaccination endotypes in single-cell RNA-sequencing uniform manifold approximation and projection (UMAP) of PBMCs from 20 healthy participants profiled by CITE-seq[10]; subsets were identified based on surface protein expression (average dsb normalized protein expression within each cluster). b,** Single-cell CITE-seq deconvolution of inflammatory genes, identified as being associated with vaccine-induced antibody response by the unsupervised and supervised approaches, in the blood immune cell subsets. The color represents average log normalized expression within each cluster with scales clipped at a maximum of 0.25, and the dot size represents the percentage of cells within that cluster with nonzero expression of the gene. HSC, hematopoietic stem cells; mDC, myeloid dendritic cell; pDC, plasmacytoid DC.

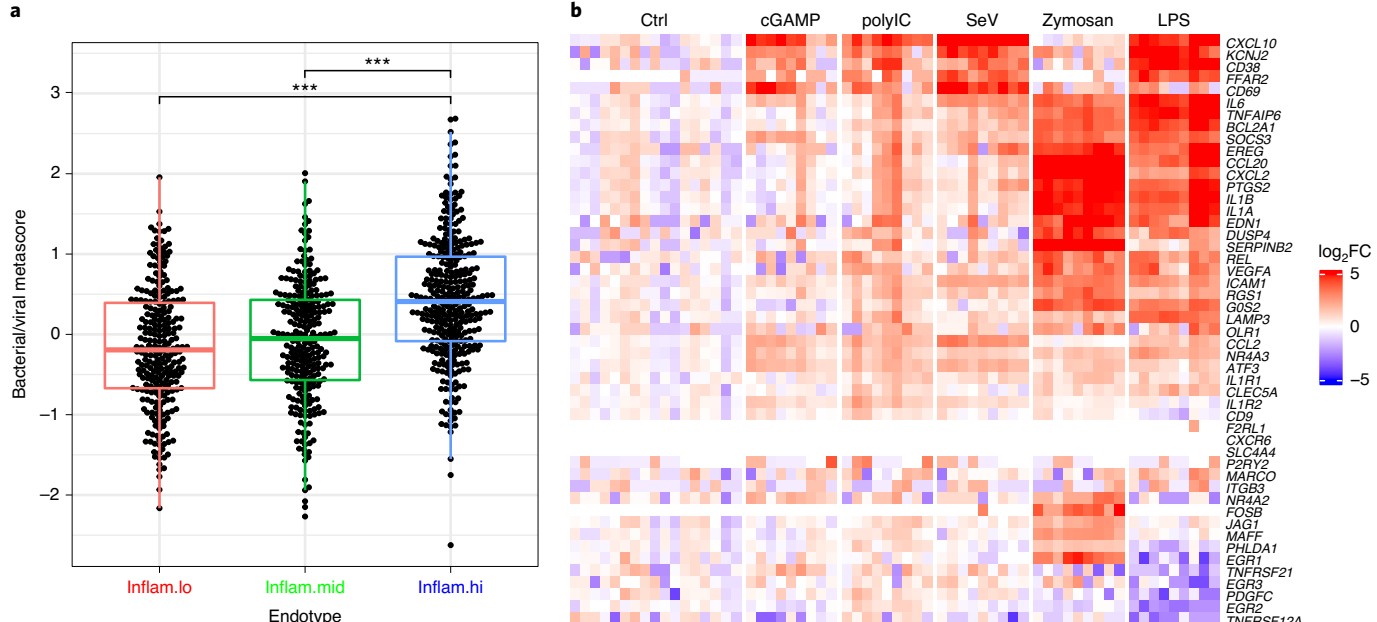

**Fig. 6 | Etiology of the pre-vaccination endotypes. a,** Box plot showing the bacterial/viral metascore as a function of the pre-vaccination inflammatory endotypes (inflam.lo, $n = 241$; inflam.mid, $n = 249$; inflam.hi, $n = 317$). For each boxplot, the vertical line indicates the median, the box indicates the interquartile range, and the whiskers indicate 1.5 times the interquartile range. A Wilcoxon rank-sum test without correction for multiple testing was used to assess the difference in bacterial/viral metascore between two endotypes: *$P < 0.05$, **$P < 0.01$ and ***$P < 0.001$. **b,** Gene expression of the inflammatory genes, identified as being associated with antibody response by the unsupervised and supervised approaches, in DCs from three independent donors stimulated for 6 h with five PRR ligands.

and heightened frequency of CD8⁺ T cells. Conversely, these same participants showed low levels of NF-κB and IRF-7 expression. In addition, the inferred frequency of CD8⁺ T cells from a deconvolution analysis was negatively correlated with day-28 antibody response, suggesting that participants of the inflam.lo states may have an activated/committed CD8⁺ T cell program before vaccination.

These two pathways could plausibly be driven by acute responses to exposure to subclinical levels of bacterial (NF-κB) or viral (interferons) infections. Genes downstream of the NF-κB and IRF-7 transcription factors were both associated with antibody responses to vaccines.

This suggests that the activation of the transcriptomic programs of those two transcription factors in innate immune cells before vaccination could lead to a more efficient priming of innate immune responses. Indeed, both interferons, TNF and the inflammasome are potent inducers of adaptive immune responses and are triggered by alum and MF59, two adjuvants that are widely used in vaccines. Of note, presence of the interferon signature before vaccination was negatively associated with the antibody responses in live viral vaccines in some populations (yellow fever, smallpox and dengue vaccine[32]). This inhibitory effect of interferons is most likely due to their antiviral

activity, which could limit viral replication and antigen presentation by these vaccines.

The transcriptomic profile of individuals in the inflam.hi state was stable over a 2-month period. This pre-vaccination inflammation could result from (1) host genetics[4], (2) the environment, for example, diet and previous infections[33] or (3) the microbiome. To the latter point, our previous work showed that TLR5-mediated sensing of flagellin in the gut microbiota promoted influenza vaccine-specific antibody response by stimulating lymph node macrophages to produce plasma cell growth factors[34]. Although we observed differences in immune cell subset frequencies between the pre-vaccination states, those frequencies could not solely explain the differences in gene expression observed between the pre-vaccination states, highlighting that in addition to differences in the cellular composition of blood, pre-vaccination states also reflect differential transcriptomic activities.

The inflammatory signature identified here was not predictive of the magnitude of the humoral response to influenza, hepatitis B and varicella zoster vaccines in older people, suggesting that age-associated inflammation[6] is different. Indeed, the inflammation signature associated with poor responses to vaccines in older people does not show overlapping genes with the inflam.hi signature associated with vaccination response in adults (18 to 55 years). Information on comorbidities and medications was not available and may constitute a confounder when comparing vaccine response in adults to that in the older population. Even so, different types of inflammation could lead to different responses to vaccination. Indeed, we provide direct evidence that distinct processes could drive diverse inflammation profiles across individuals.

Strategies that directly impact pre-vaccination inflammation or modulate the pre-vaccination commensal bacterial flora impact the immune response to vaccination[15,27]. In this study, we observed similarities between the pro-inflammatory signature associated with vaccine response and the pro-inflammatory signatures induced by bacterial infections. The latter activate pattern-recognition receptor signaling cascades, which will trigger the activation of the NF-κB transcription factor complex and the induction of pro-inflammatory transcriptomic programs including pro-inflammatory cytokines (for example, IL-1). The overlap between the pro-inflammatory signatures associated with vaccine response and that following bacterial signaling was not specific to one bacterial species but was shared by different bacteria such as *S. aureus* and *S. pneumoniae*. Importantly, these signatures overlapped with that of the activation by PRR ligands of bacterial (TLR1, TLR2 and TLR4) or viral (polyIC) pathogens. Among the 13 vaccines that are part of the Immune Signatures Data Resource, only the hepatitis B vaccine was adjuvanted with aluminum hydroxide. The other vaccines did not use an adjuvant and having a pro-inflammatory signature pre-vaccination provides an activated innate immune state with overlap with states induced by adjuvant and could explain the association with an enhanced humoral response after vaccination. These findings are of even greater relevance as the quest to develop durable and efficacious vaccine platforms for severe acute respiratory sydrome coronavirus 2 (SARS-CoV-2) have become a global health priority. Identifying adjuvants that will enable the different SARS-CoV-2 vaccine platforms will benefit from our findings.

In conclusion, we have identified an inflammatory signature downstream of transcription factors NF-kB and IRF-7 in innate immune cells that predicts humoral response across diverse vaccines. This provides a mechanistic framework that can lead to the selection of adjuvants most efficient at stimulating vaccine-induced protective immune responses.

## Online content

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

[1]Department of Pathology and Laboratory Medicine, Emory University, Atlanta, GA, USA. [2]Center for Biostatistics, Icahn School of Medicine at Mount Sinai, New York, NY, USA. [3]Multiscale Systems Biology Section, Laboratory of Immune System Biology, NIAID and Center for Human Immunology (CHI), NIH, Bethesda, MD, USA. [4]NIH-Oxford-Cambridge Scholars Program, Cambridge University, Cambridge, UK. [5]Yale School of Medicine, New Haven, CT, USA. [6]Division of Transplant Surgery, University of California, San Francisco, San Francisco, CA, USA. [7]Fred Hutchinson Cancer Research Center, Seattle, WA, USA. [8]Stanford University School of Medicine, Stanford University, Stanford, CA, USA. [9]Precision Vaccines Program, Boston Children's Hospital, Boston, MA, USA. [10]Harvard Medical School, Boston, MA, USA. [11]ImmPort Curation Team, NG Health Solutions, Rockville, MD, USA. [12] Biomedical Data Science Center, University of Lausanne and Lausanne University Hospital, Lausanne, Switzerland. [13]Swiss Institute of Bioinformatics, Lausanne, Switzerland. [22]These authors jointly supervised this work: Steven H. Kleinstein, Rafick-Pierre Sékaly. *A list of authors and their affiliations appears at the end of the paper. ✉e-mail: steven.kleinstein@yale.edu; rafick.sekaly@emory.edu

**The Human Immunology Project Consortium (HIPC)**

**Alison Deckhut-Augustine**[14], **Raphael Gottardo**[7,12,13], **Elias K. Haddad**[15], **David A. Hafler**[5], **Eva Harris**[16], **Donna Farber**[17], **Steven H. Kleinstein**[5,22], **Ofer Levy**[9,10], **Julie McElrath**[7], **Ruth R. Montgomery**[7], **Bjoern Peters**[18], **Bali Pulendran**[8], **Adeeb Rahman**[19], **Elaine F. Reed**[20], **Nadine Rouphael**[1], **Minnie M. Sarwal**[6], **Rafick-Pierre Sékaly**[1,22], **Ana Fernandez-Sesma**[19], **Alessandro Sette**[18], **Kenneth D. Stuart**[21], **Alkis Togias**[14] and **John S. Tsang**[3]

[14]NIAID, NIH, Bethesda, MD, USA. [15] Department of Medicine, Drexel University, Philadelphia, PA, USA. [16] School of Public Health, University of California, Berkeley, Berkeley, CA, USA. [17]Columbia University Medical Center, New York, NY, USA. [18]La Jolla Institute for Immunology, La Jolla, CA, USA. [19]Icahn School of Medicine at Mount Sinai, New York, New York, NY, USA. [20]David Geffen School of Medicine at University of California, Los Angeles, CA, USA. [21]Seattle Children's Research Institute, Seattle, WA, USA.

## Methods

### Gene expression preprocessing

ImmPort (release June 2022) and ImmuneSpace (release December 2021) software were used to collect the transcriptomic and phenotypic data. An extensive description of the preprocessing of microarray and RNA-sequencing datasets included in the Immune Signatures Data Resource can be found in ref. [16]. The dataset includes 2,949 samples from published studies and 228 samples not included in previously published studies. Those 2,949 samples originate from 820 participants, 800 reported as healthy and 20 (<3%) with type 2 diabetes. All these samples were assembled into a single resource. Briefly, raw probe intensity data for Affymetrix studies were background corrected and summarized using the RMA algorithm implemented in R (version 4.2.0) and Bioconductor (version 3.14). For studies using the Illumina array platform, background-corrected raw probe intensities were used. Expression data within each study were quantile normalized and log transformed separately for each study.

### Batch correction

An extensive description of the across-studies normalization used to correct for batch effects can be found in ref. [16]. Briefly, a linear model was fit using the pre-vaccination normalized gene expression as a dependent variable and platform, study and blood sample type (that is, whole blood or PBMCs) as independent variables. The estimated effect of the platform, study and sample type was then subtracted from the entire gene expression (before and after vaccination) to obtain the batch-corrected gene expression used for the analysis presented herein. Principal variance component analysis was used to assess the effect of other phenotypic variables on the batch-corrected gene expression[35]. All the phenotypic variables were coded as categorical variables before the principal variance component analysis; this included the imputed age coded as 10-year intervals and the time points before and after vaccination, which were left censored at 20 d and coded as days from vaccination.

### Clustering of the samples

For functional characterization of the genes, we made use of known gene sets from two sources: Hallmark collection from MSigDB (version 7.2)[17] and the BTMs[18]. Overall activity of each gene set/pathway was estimated for each sample using SLEA[36]. Hierarchical clustering using Euclidean distance and complete linkage was used to cluster samples. The resulting dendrogram was cut to generate three clusters of samples. The three clusters were designated as inflam.lo, inflam.mid and inflam.hi, based on the average SLEA z-score of four hallmark inflammatory gene sets (HALLMARK_INFLAMMATORY_RESPONSE, HALLMARK_COMPLEMENT, HALLMARK_IL6_JAK_STAT3_SIGNALING and HALLMARK_TNFA_SIGNALING_VIA_NFKB). Hallmark and BTM gene sets were grouped based on their name and description into markers of seven cell subsets or canonical pathways (T cells, NK cells, B cells, monocytes/DCs, inflammation, E2F/MYC and ISGs). Canonical genes of those seven cell subsets or canonical pathways were identified by looking at the genes part of the gene sets annotated to those cell subsets or canonical pathways and ranking them based on the number of GeneRIF entries associating them to cell subsets or canonical pathways.

### Antibody response

Because some vaccines include multiple strains of viral antigens, the fold change in the response metric was defined as the MFC of any strain in the vaccine at day 28 (± 2 d) compared to before vaccination. MFC was calculated for all participants with neutralizing antibody response, hemagglutination inhibition, or IgG levels measured by ELISA[16].

### Identification of high and low responders

The MFC between day 28 (± 2 d) and pre-vaccination titers was used to quantify the antibody response to vaccination. To minimize the difference in antibody response between studies (for example, due to different vaccines or different techniques used for antibody concentration assessment), the high and low responders were identified for each study separately by selecting the participants with an MFC value equal or above the 70th percentile as high responders and participants with an MFC value equal or below the 30th percentile as low responders.

### Strategy to identify signature predictive of vaccine response

To evaluate if participant-specific transcriptomic profiles taken before vaccination were predictive of antibody response 28 d after vaccine, we developed predictive models using the random forest algorithm. The training set included participants achieving a high or low antibody response ($n$ = 522) based on the discretization of MFC (MFC_p30) and the top 500 varying genes as features (based on variance calculated across all pre-vaccination samples part of the Immune Signatures Data Resource with available antibody-response data). The predictive model was trained to maximize the auROC, and tuning parameters were estimated using tenfold cross-validation. In this final model, the performance was assessed using tenfold cross-validation with standard performance metrics including auROC, accuracy, positive predictive value, negative predictive value, sensitivity, specificity, as well as Brier score.

MetaIntegrator[37] (version 2.1.3) was used to apply previously identified pre-vaccination signatures of vaccine response and the most important genes in the classifier identified in this work (importance > 50%) to the different studies of the Immune Signatures Data Resource. The auROC was used to assess the accuracy of the signatures. Significance was assessed by comparing the observed auROC against a background distribution generated from permuted high-response and low-response labels.

### CITE-seq analysis

CITE-seq data consisting of pre-vaccination PBMC samples from healthy participants were downloaded from ref. [10]. Cell type annotations used in this analysis were the 'high-resolution' annotations from work by Kotliarov et. al., and are based on graph-based clustering using Seurat[38] directly on a Euclidean distance matrix of surface protein expression. CITE-seq surface protein data were normalized and denoised using dsb R package[39]. UMAP embeddings[40] were calculated using Seurat. Presto[41] was used to generate a rank list of genes most specific to each cell type using 18,997 genes expressed in a minimum of 5 cells based on a one-cell type vs all other Wilcoxon tests, and gene set enrichment analysis of the predictive signature against this rank list was tested using the fgsea package[42].

### Comparison with viral/bacterial signatures

The bacterial/viral classifier was applied to the immune signature dataset by averaging the expression of the bacterial infection markers (*HK3*, *TNIP1*, *GPAA1* and *CTSB*) and subtracting the average expression of the viral infection markers (*IFI27*, *JUP* and *LAX1*); a resulting score above or equal to 0 was considered more similar to bacterial infection, while a score below 0 was considered more similar to viral infection.

### Statistical analysis

Association between categorical variables was assessed using Fisher's exact test. Association between a categorical and a continuous variable was assessed using a Kruskal–Wallis and Wilcoxon rank-sum test. Association between continuous variables was assessed using a Spearman correlation and *t*-test. *P* values were adjusted for multiple testing using the Benjamini–Hochberg correction.

### Reporting summary

Further information on research design is available in the Nature Research Reporting Summary linked to this article.

## Data availability

All data used in this study are available from ImmuneSpace (www.immunespace.org/is2.url).

## Code availability

R code used to generate the figures presented in the paper can be found at ImmuneSpace.

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

## Acknowledgements

This research was performed as a project of the Human Immunology Project Consortium (HIPC) and supported by the National Institute of Allergy and Infectious Diseases (NIAID). This work was supported in part by National Institutes of Health (NIH) grants U19AI118608 (to O.L.), U19AI128949 and U19AI090023 (to B.P.), U19AI118626 and U19AI089992 (to S.H.K.), U19AI128914 and U19AI128910 (to R.-P.S.), U19AI118610 (to M.S.-F.) and U19AI128913, and the Intramural Program of NIAID and NIH institutes supporting the Trans-NIH Center for Human Immunology.

## Author contributions

S.F. and S.H.K. performed conception or design of the work. Analysis was completed by S.F., L.E.T., M.P.M., D.G.C., B.G. and D.R. Acquisition was conducted by E.H., H.E.R.M., T.H., J.D.-A., P.D., HIPC, O.L., R.G., M.M.S., J.S.T. and M.S.-F. S.F., B.P., S.H.K. and R.-P.S performed interpretation of the data and drafted the work.

## Competing interests

O.L. is a named inventor on patents held by Boston Children's Hospital regarding human in vitro systems modeling vaccine action and vaccine adjuvants. B.P. serves on the External Immunology Network of GSK, and on the scientific advisory board of Medicago, CircBio, Sanofi, EdJen and Boehringer-Ingelheim. S.H.K. receives consulting fees from Northrop Grumman and Peraton. T.H. owns stock in GSK and Pfizer. The remaining authors declare no competing interests.

## Additional information

**Extended data** is available for this paper at https://doi.org/10.1038/s41590-022-01329-5.

**Correspondence and requests for materials** should be addressed to Steven H. Kleinstein or Rafick-Pierre Sékaly.

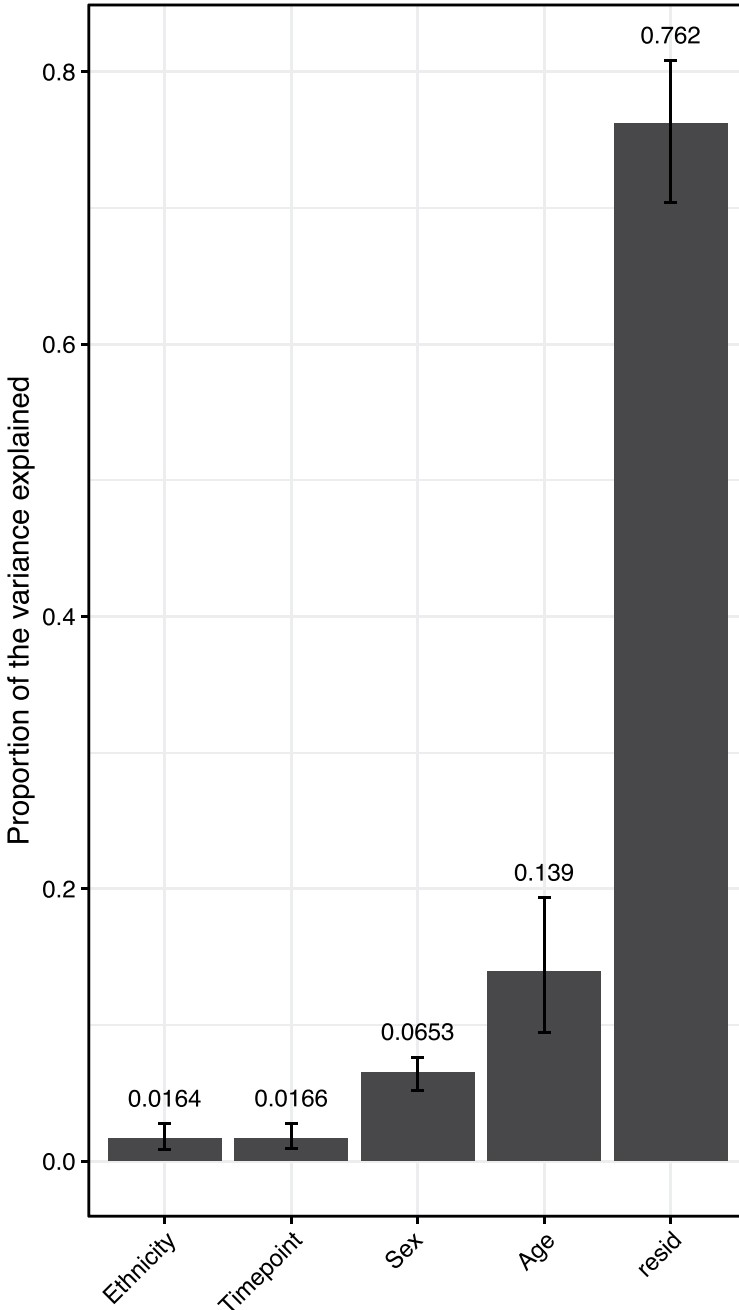

**Extended Data Fig. 1 | Principal variance component analysis using pre-vaccination transcriptomic expression.** All phenotypic variables were coded as categorical variables. The variance explained by each variable (x-axis) is indicated as the label at the top of each bar. 95% intervals of confidence were calculated by bootstrapping the samples (4000 bootstrap iterations). resid: residuals.

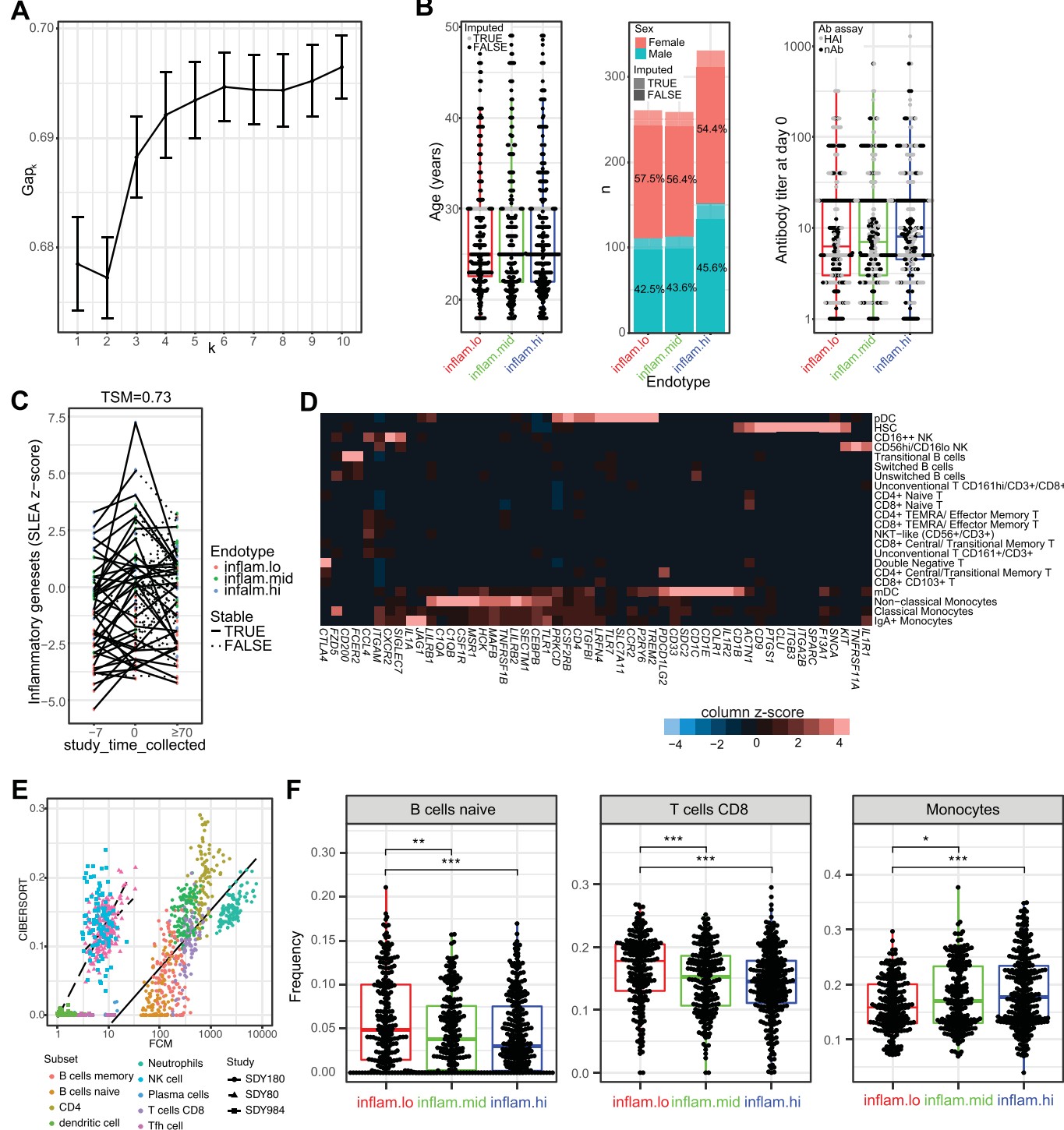

**Extended Data Fig. 2 | Identification of the pre-vaccination endotypes. (a)**
Gap statistic for different number of clusters (k). The 95% confidence interval
were estimated by 100 Monte Carlo iterations. **(b)** Boxplot and barplot of age, sex
and pre-vaccination antibody titers as a function of the inflammatory endotypes.
For each boxplot, the vertical line indicates the median, the box indicates the
interquartile range, and the whiskers indicate 1.5 times the interquartile range.
**(c)** Lineplot showing the differences in expression of inflammatory pathways
between 7 days before vaccination compared to just before vaccination. **(d)**
Monocytes/dendritic cell markers expression in blood of pre-vaccinated
individuals. PBMCs from 20 healthy participants profiled by CITE-seq[10].
The heatmap shows the average expression of monocyte and dendritic cell

markers identified in bulk meta-analysis associated with the high inflammatory
state, scale shown is the z-score of the gene across protein-based cell subsets.
**(e)** Scatter plot showing inferred frequencies of immune cells estimate by
CIBERSORT as a function of cell counts measured by flow cytometry (FCM) for
three separate study. Linear fit (lines) is drawn for each study. **(f)** Frequencies of
immune cells, estimated by deconvolution, in the three pre-vaccination states.
For each boxplot, the vertical line indicates the median, the box indicates the
interquartile range, and the whiskers indicate 1.5 times the interquartile range.
Wilcoxon rank-sum test between two endotypes: p-values less than 0.05 are
flagged with one star (*), p-values less than 0.01 are flagged with 2 stars (**), and
p-values less than 0.001 are flagged with three stars (***).

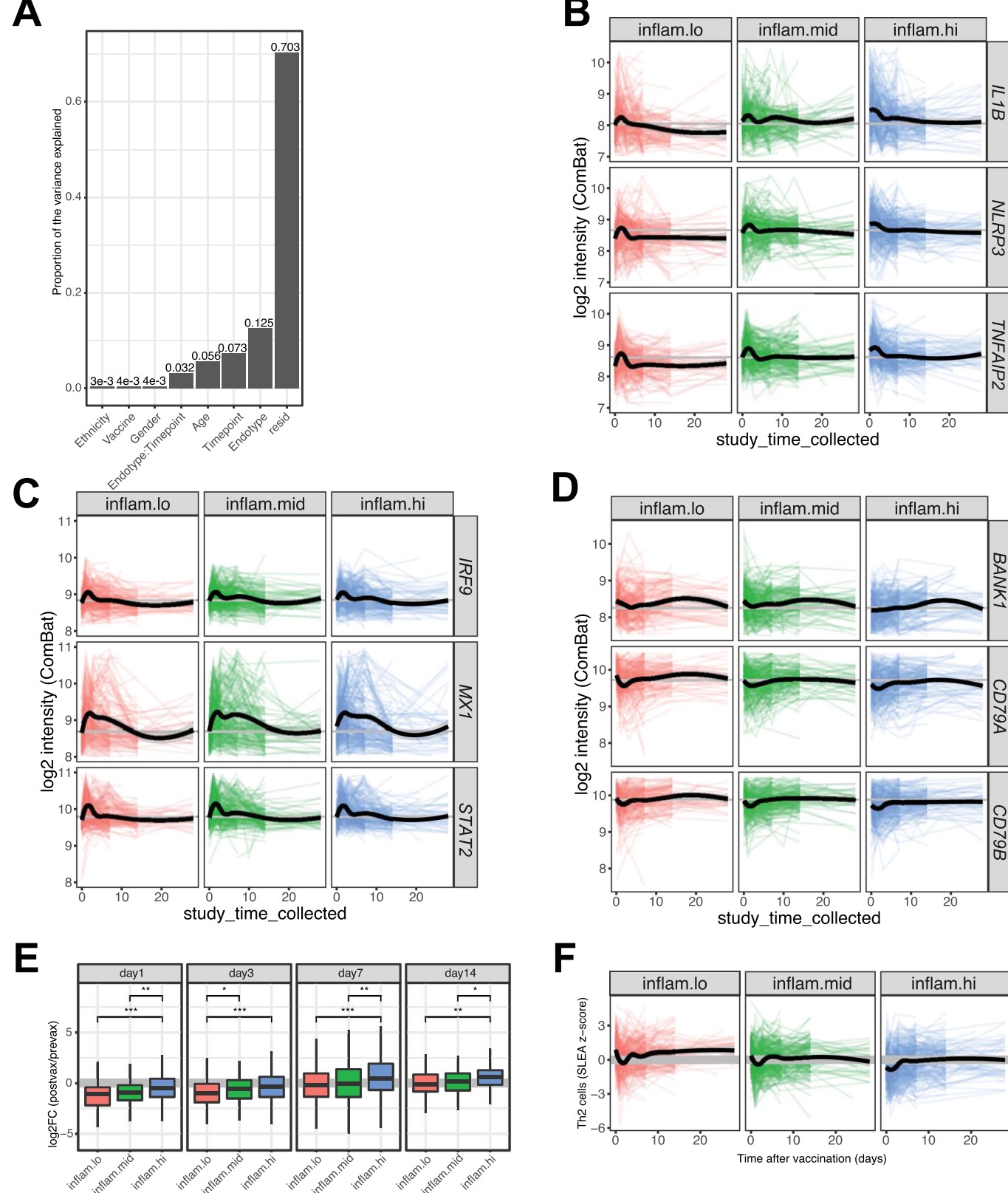

**Extended Data Fig. 3 | Pre-vaccination endotypes affect post-vaccination transcriptomic response. (a)** Principal variance analysis with the inflammatory states. Canonical inflammatory genes **(b)**, interferon-stimulated genes **(c)**, and B cell markers **(d)** expression over time in the three inflammatory states. **(e)** Log2 fold-change over pre-vaccination levels of B cell markers. For each boxplot, the vertical line indicates the median, the box indicates the interquartile range, and the whiskers indicate 1.5 times the interquartile range. Wilcoxon rank-sum test between two endotypes: p-values less than 0.05 are flagged with one star (*), p-values less than 0.01 are flagged with 2 stars (**), and p-values less than 0.001 are flagged with three stars (***). **(f)** Th2 cell markers over time.

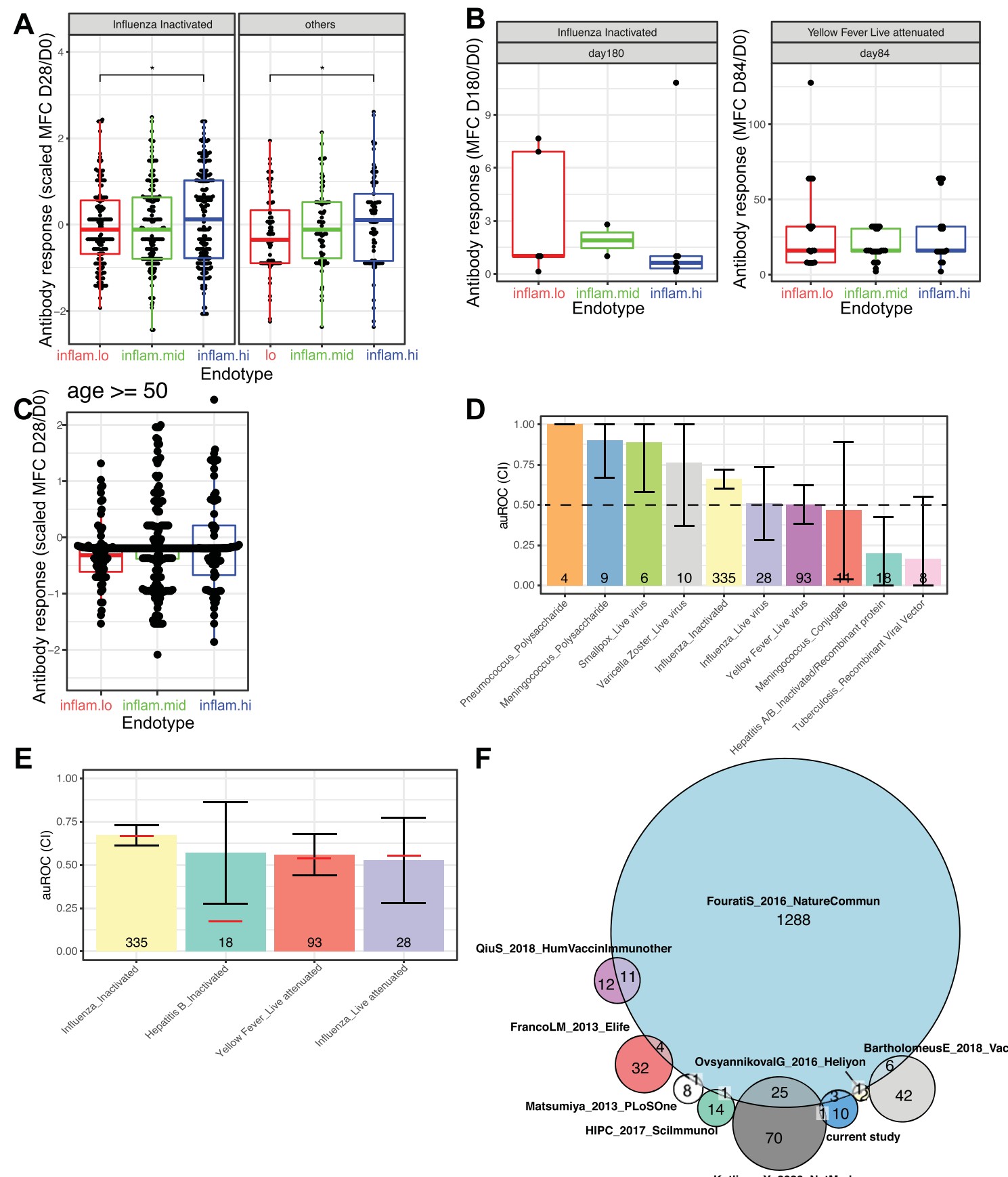

**Extended Data Fig. 4 | See next page for caption.**

**Extended Data Fig. 4 | Pre-vaccination endotypes predict antibody response to vaccines. (a)** Association between the unsupervised pre-vaccination cluster and antibody response at day 28 for (right) Influenza inactivated vaccines and (left) other vaccines. For each boxplot, the vertical line indicates the median, the box indicates the interquartile range, and the whiskers indicate 1.5 times the interquartile range. Wilcoxon rank-sum test between two endotypes: p-values less than 0.05 are flagged with one star (*), p-values less than 0.01 are flagged with 2 stars (**), and p-values less than 0.001 are flagged with three stars (***). **(B)** Association between the unsupervised pre-vaccination cluster and antibody response at day 180 for the inactivated influenza vaccine (left) and day 84 yellow fever vaccine (right). For each boxplot, the vertical line indicates the median, the box indicates the interquartile range, and the whiskers indicate 1.5 times the interquartile range. **(c)** Association between the unsupervised pre-vaccination cluster and antibody response at day 28 for healthy adults aged 50 and above. For each boxplot, the vertical line indicates the median, the box indicates the interquartile range, and the whiskers indicate 1.5 times the interquartile range. **(d)** Accuracy of the supervised classifier to predict the antibody response groups per vaccine. Mean and 95% CI for auROC across the 10-folds for each vaccine separately. **(e)** Accuracy of supervised classifiers trained on a specific vaccine and tested on the same vaccine by cross-validation. The red line indicates the accuracy of the pan-vaccine classifier while the bar represents the 95% CI calculated by 10-fold cross-validation. **(f)** Venn diagram of the overlap of inflammatory genes and previously identified pre-vaccination signature of vaccine response.

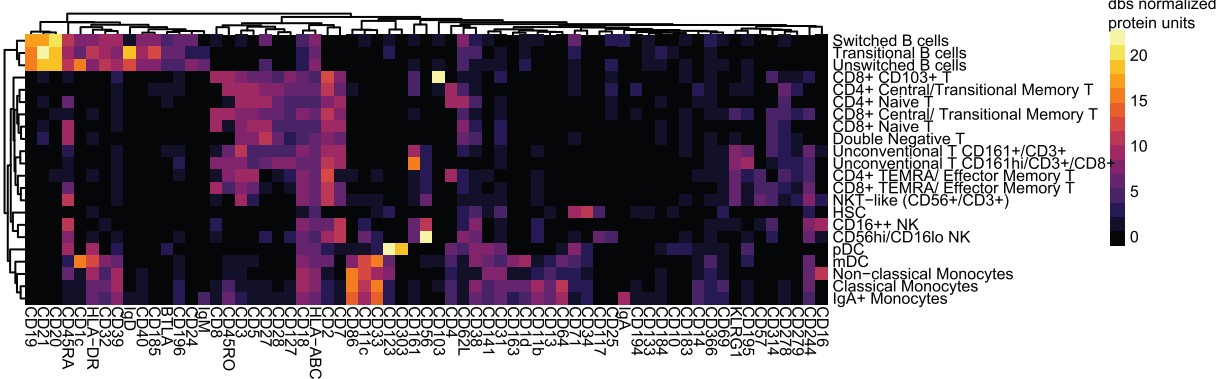

**Extended Data Fig. 5 | PBMCs from 20 healthy participants profiled by CITE-seq.** Subsets were identified based on surface protein expression (average dsb normalized protein expression within each cluster).

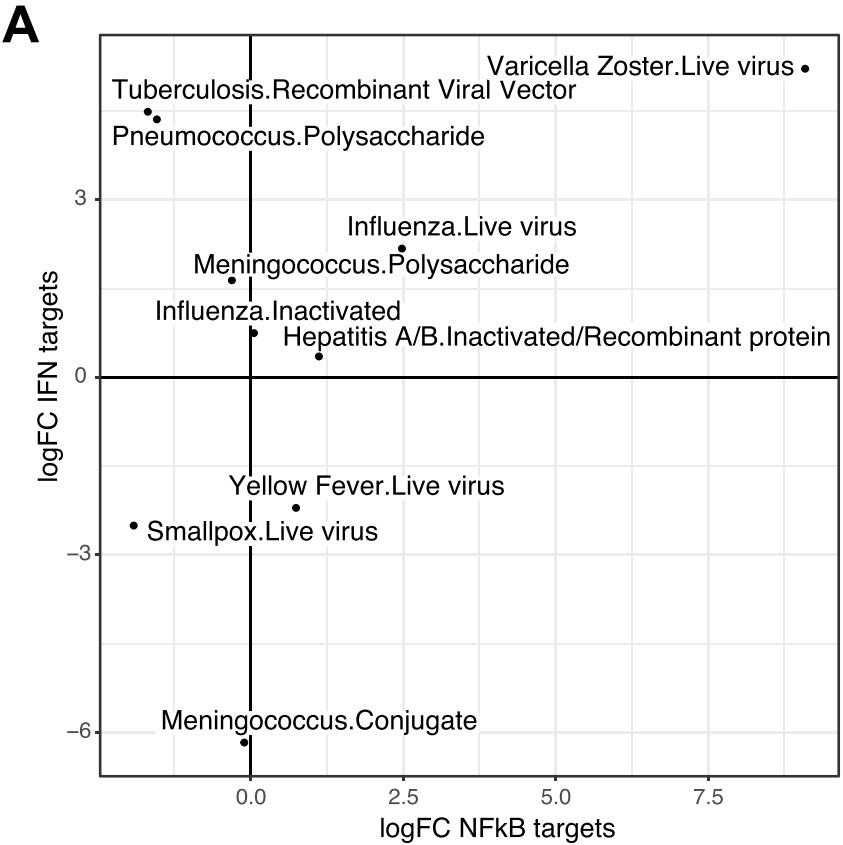

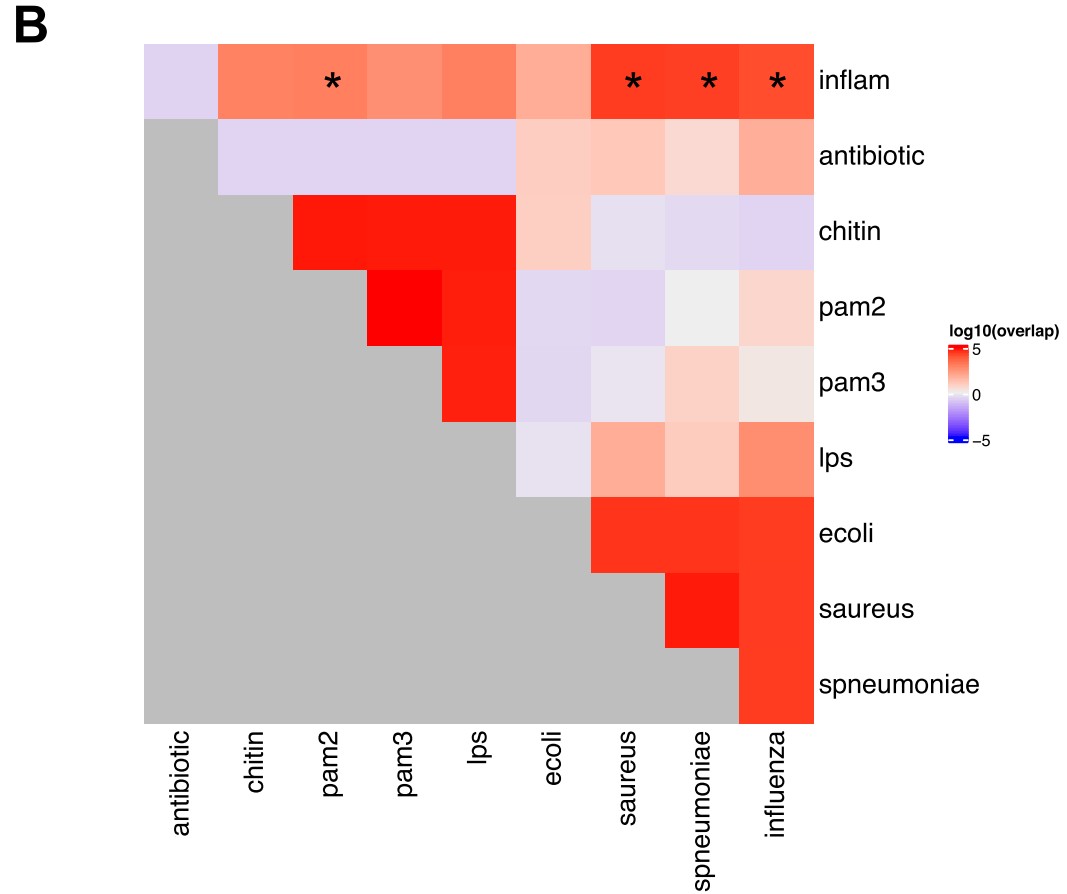

**Extended Data Fig. 6 | See next page for caption.**

**Extended Data Fig. 6 | Upstream signaling molecules and transcription factors demarcate the pre-vaccination endotypes. (a)** Scatter plot showing the discriminative power of NFκB- and Interferon-target genes for each of the vaccine in the 'Immune Signatures Data Resource'. LogFC of the SLEA z-score of the two genesets between high vaccine responders and low vaccine responders is shown. **(b)** Overlap between the genes differentially expressed between the inflam.hi and inflam.lo endotypes and inflammatory signatures described in the literature. The significance of the overlap between ranked lists of genes was assessed by permutation. * indicate statistically significant overlap (permutation test: $p \leq 0.05$) between the differentially expressed genes between the inflam.hi and inflam.lo endotypes and previously described inflammatory signatures extracted from the literature.

# Reporting Summary

## Statistics

For all statistical analyses, confirm that the following items are present in the figure legend, table legend, main text, or Methods section.

| n/a | Confirmed | |
|---|---|---|
| ☐ | ☒ | The exact sample size (*n*) for each experimental group/condition, given as a discrete number and unit of measurement |
| ☐ | ☒ | A statement on whether measurements were taken from distinct samples or whether the same sample was measured repeatedly |
| ☐ | ☒ | The statistical test(s) used AND whether they are one- or two-sided *Only common tests should be described solely by name; describe more complex techniques in the Methods section.* |
| ☐ | ☒ | A description of all covariates tested |
| ☐ | ☒ | A description of any assumptions or corrections, such as tests of normality and adjustment for multiple comparisons |
| ☐ | ☒ | A full description of the statistical parameters including central tendency (e.g. means) or other basic estimates (e.g. regression coefficient) AND variation (e.g. standard deviation) or associated estimates of uncertainty (e.g. confidence intervals) |
| ☐ | ☒ | For null hypothesis testing, the test statistic (e.g. *F*, *t*, *r*) with confidence intervals, effect sizes, degrees of freedom and *P* value noted *Give P values as exact values whenever suitable.* |
| ☒ | ☐ | For Bayesian analysis, information on the choice of priors and Markov chain Monte Carlo settings |
| ☐ | ☒ | For hierarchical and complex designs, identification of the appropriate level for tests and full reporting of outcomes |
| ☒ | ☐ | Estimates of effect sizes (e.g. Cohen's *d*, Pearson's *r*), indicating how they were calculated |

*Our web collection on statistics for biologists contains articles on many of the points above.*

## Software and code

Policy information about availability of computer code

| Data collection | ImmPort (software release June 2022) and ImmuneSpace (software release December 2021) were used to collect the transcriptomic and phenotypic data used in the manuscript. |
|---|---|
| Data analysis | R (version 4.2.0) and Bioconductor (version 3.14) were used to generate the figures of the manuscript. MSigDB (version 7.2) was used for pathway analysis. MetaIntegrator (version 2.1.3) was used to test previous pre-vaccination signatures. UMAP of the umap python package (version 0.1.1) was used for dimension reduction of the scRNASeq data. |

For manuscripts utilizing custom algorithms or software that are central to the research but not yet described in published literature, software must be made available to editors and reviewers. We strongly encourage code deposition in a community repository (e.g. GitHub). See the Nature Portfolio guidelines for submitting code & software for further information.

## Data

Policy information about availability of data

All manuscripts must include a data availability statement. This statement should provide the following information, where applicable:
- Accession codes, unique identifiers, or web links for publicly available datasets
- A description of any restrictions on data availability
- For clinical datasets or third party data, please ensure that the statement adheres to our policy

The data used in the paper can be found at www.immunespace.org/is2.url

# Human research participants

Policy information about studies involving human research participants and Sex and Gender in Research.

| | |
|---|---|
| Reporting on sex and gender | Participants of all sexes were used in this study. For participants with missing sex, mRNA expression of genes on the Y chromosome was used to infer sex. |
| Population characteristics | Healthy participants agesd 18 years and above were included. |
| Recruitment | Not applicable. A meta-analysis of the publicly available data was performed in this manuscript. |
| Ethics oversight | Not applicable. No recruitment or clinical trial were performed in this manuscript. |

Note that full information on the approval of the study protocol must also be provided in the manuscript.

# Field-specific reporting

Please select the one below that is the best fit for your research. If you are not sure, read the appropriate sections before making your selection.

☒ Life sciences ☐ Behavioural & social sciences ☐ Ecological, evolutionary & environmental sciences

For a reference copy of the document with all sections, see nature.com/documents/nr-reporting-summary-flat.pdf

# Life sciences study design

All studies must disclose on these points even when the disclosure is negative.

| | |
|---|---|
| Sample size | no simple size calculation was performed. All available transcriptomic data related to vaccine response available on December 2020 was used in this manuscript. |
| Data exclusions | Transcriptomic data on children and juvenile (age < 18 years) was excluded from this analysis. Only human data was included. Those two exclusion criteria were pre-established. |
| Replication | 10-fold cross-validation was used to assess the reproducibility of the classifier generated in the manuscript. |
| Randomization | No randomization was performed. All available samples were used to perform the meta-analysis. No potential confounders were identified using principal variance component analysis (PVCA). |
| Blinding | Blinding of the response data was not possible because this meta-analysis rely on publicly available data. |

# Reporting for specific materials, systems and methods

We require information from authors about some types of materials, experimental systems and methods used in many studies. Here, indicate whether each material, system or method listed is relevant to your study. If you are not sure if a list item applies to your research, read the appropriate section before selecting a response.

## Materials & experimental systems

| n/a | Involved in the study |
|---|---|
| ☒ | ☐ Antibodies |
| ☒ | ☐ Eukaryotic cell lines |
| ☒ | ☐ Palaeontology and archaeology |
| ☒ | ☐ Animals and other organisms |
| ☒ | ☐ Clinical data |
| ☒ | ☐ Dual use research of concern |

## Methods

| n/a | Involved in the study |
|---|---|
| ☒ | ☐ ChIP-seq |
| ☒ | ☐ Flow cytometry |
| ☒ | ☐ MRI-based neuroimaging |

