## [Peer Review File · Nature Immunology]

Peer Review Information

Journal: Nature Immunology

Manuscript Title: Pan-vaccine analysis reveals innate immune endotypes predictive of antibody responses to vaccination

Corresponding author name(s): Rafick Sekaly and Steven H. Kleinstein

Reviewer Comments & Decisions:

Decision Letter, initial version:
--

Subject: Decision on Nature Immunology submission NI-A33975

Message: 18th May 2022

Dear Dr Sekaly,

Your Article, "Pan-vaccine analysis reveals innate immune endotypes predictive of antibody responses to vaccination" has now been seen by 3 referees. You will see from their comments below that while they find your work of interest, some important points are raised. We are very interested in the possibility of publishing your study in Nature Immunology, but would like to consider your response to these concerns in the form of a revised manuscript before we make a final decision on publication.

In particular, it has been raised that it would be important to measure antibody levels.

We therefore invite you to revise your manuscript taking into account all reviewer and editor comments. Please highlight all changes in the manuscript text file in Microsoft Word format.

* If you have not done so already please begin to revise your manuscript so that it conforms to our Article format instructions at

<http://www.nature.com/ni/authors/index.html>. Refer also to any guidelines provided in this letter.

* Please include a revised version of any required reporting checklist. It will be available to referees to aid in their evaluation of the manuscript goes back for peer review. They are available here:

Reporting summary:

Please use the link below to submit your revised manuscript and related files:
[REDACTED]

We hope to receive your revised manuscript within two weeks but if you require more time, especially to add antibody data please reach out.

Nature Immunology is committed to improving transparency in authorship. As part of our efforts in this direction, we are now requesting that all authors identified as 'corresponding author' on published papers create and link their Open Researcher and Contributor Identifier (ORCID) with their account on the Manuscript Tracking System (MTS), prior to acceptance. ORCID helps the scientific community achieve unambiguous attribution of all scholarly contributions. You can create and link your ORCID from the home page of the MTS by clicking on 'Modify my Springer Nature account'. For more information please visit www.springernature.com/orcid.

Sincerely,

Jamie D.K. Wilson, D.Phil
Chief Editor
Nature Immunology
212 726 9207
j.wilson@us.nature.com

Reviewers' Comments:

Reviewer #1:

Remarks to the Author:

Fourati and colleagues analyzed peripheral blood transcriptional profile of a cohort of 820 subjects (18-55y) before and after vaccination. By profiling the subjects in three categories, the authors report higher serum antibody responses in the one associated with pro-inflammatory responses at baseline. In particular, NFkB and IFN pathways activation before vaccination had a strong correlation with vaccine response. This could be linked to pre-existing inflammation caused by exposure to viruses or bacteria, also in line with previous studies suggesting a beneficial link between microbiota and immunity. Thus the authors conclude that adjuvants modulating NFkB before vaccination might improve vaccine response.

Overall, the study is well thought and, by reconciling genetic signatures with vaccines immunity, could provide information that could help guide adjuvant design and patient stratification, although experimental validation is missing.

There are several elements to be clarified:

1. The population is divided in clusters with low, medium and high level of pre-vaccination inflammatory transcripts. Can this be confirmed with evidence that pre-vaccination levels of pro-inflammatory cytokines in humans correlate with higher vaccine antibody response? Were confounding effects such as use of corticosteroids considered in the study?
2. In Figure 3, a reduced expression of pro-inflammatory genes after vaccination is observed for inflam.hi group as compared to inflam.low (including lower B cell response). What is the proposed mechanism that correlate the initial reduced immune stimulation with increase antibody response?
3. Pre-existing inflammation in the elderly is characterized by reduced vaccine immune response. The same authors reported this effect in a population mostly 65-75 resulted in lower HBV vaccine response (<https://doi.org/10.1038/ncomms10369>). How the evidence reported in this manuscript can inform vaccine design for this population?
4. The authors propose the use of adjuvants to stimulate NFkB pathway before vaccination to validate the proposed immunological mechanism. Are there evidences supporting this claim?

Reviewer #2:

Remarks to the Author:

Review for "Pan-vaccine analysis reveals innate immune endotypes predictive of responses to vaccination" by Dr. Fourati et al. The manuscript describes a unique deep analysis of a large transcriptomic dataset developed by the HIPC NIH consortium consisting of data from 28 vaccine studies including transcriptomics data on 820 individuals. Given the limited impact of demographics on shaping the vaccine induced immune response, the group found 3 distinct transcriptional signatures prior to vaccination across all vaccinees- comprising of a high, medium, and low endotypes – linked to striking differences in the pro-inflammatory status, cell proliferative activity, and metabolic state of the immune transcriptome. The signature pointed to a differential response across innate, B cells, and T cells following vaccination, was largely enriched in myeloid cells, and was linked to increased bacterial-infection signature, and could predict vaccine response across most vaccines. These data points to a novel axis of pre-existing immunity across vaccines that may prepare the foundational state of the immune system to be more responsive to vaccination. While data has suggested that alterations in myeloid cells can interfere with the response to vaccines in the past- this study flips this observation- to show that particular pre-existing myeloid signatures may be required to fully potentiate vaccine induced immunity – representing a truly innovative way to approach next generation vaccine development.

Comments:

1. A tremendous amount of work has gone in to the generation and analysis of the transcriptomic data- but immune data is not included. Specifically, the magnitude of the antibody or T cell response are missing – to substantiate that the magnitude of the transcriptomic signature is a close surrogate of the adaptive immune response.
2. Pre-vacc endotype differences explained 12.5% of variation in gene expression pre and post infection- approximating the level of variance captured by demographics alone. It would be helpful to show some supplemental data that the variance captured by both these sets of parameters are independent.
3. The B cell signatures in Figure 3 are described as "dampened". While this is clearly one interpretation- could they simply not be in the blood if there is an ongoing or recent infection? Perhaps "diminished in the periphery" could be used interchangeably? Some discussion would be helpful on this subject.
4. The T cell data in Figure 3 supplemental seems quite exciting and should be considered for the main body.
5. The relationship between the endotypes and antibody results at day 28 is buried in the supplemental data – perhaps as it did not reach statistical significance – but represents one of the only analyses that links directly to antibodies. It would be helpful to include this in the main body and any other analyses that directly relate the immunological endpoint of greatest interest in vaccine development (antibody titers).
6. The differences in endotype prediction across vaccines is fascinating- particular in the context of particular vaccines. Were vaccines grouped into vectors, polysaccharides, those to which pre-existing immunity exists, etc. to enhance power and explore endotype predictive accuracy on response profiles? Some discussion/explanation would be very helpful to understand what is common about the vaccines for which the endotypes are highly predictive??? Are they more IFN dependent? Are they novel responses?
7. Figure 6a shows the program in the "high" endotype – which is striking. It would be helpful to see the other endotypes as well in supplemental- to visualize the universality of

these profiles- that presumably would be present irrespective of the level of viral/bacterial exposure.

8. It is unclear whether the viral vs bacterial signatures relate to pathogens or commensals? Are the LPS/zymosan signatures related to pathogenic bacteria or simply to microbial translocation and/or microbiome shifts? Where signatures of microbial translocation also explored and does this explain this bacterial signature? This would be critical to explore/add. Presumably, cytokine/serum analytes were also captured in the HIPC and could add tremendous value to understanding this baseline advantage in the high endotype population.

9. While antibodies represent the surrogate of protection against most infections, some discussion on whether these are mechanistic correlates and whether these mark durability differences should be mentioned.

Reviewer #3:

None

Reviewer #4:

Remarks to the Author:

This manuscript aims to show how pre-existing immune conditions, determined by blood transcriptomic profiling, influence the immune response to a set of 13 different vaccines in 820 recipients. Data from 28 studies are combined. The vaccines included both live attenuated and inactivated immunogens.

The main findings are that the pre-vaccination group could be divided into three endotypes based on inflammation markers. Those with a strong pre-existing pro-inflammatory gene expression, all involving NFkB regulation, showed stronger antibody responses after vaccination. However the differences were not large. The scale on the y axis is not clear in Fig 4A, I assume it is on a log2 scale. I am not expert in interpreting ROC plots but the prediction doesn't look particularly strong. However when broken down for the inactivated influenza vaccine the p value was very impressive. The heterogeneity of some of the cohorts, the different types of vaccine and small numbers in some parts of the study may have lessened the overall effect. The strong proinflammatory response was similar to that stimulated by TLR ligands/adjuvants and is connected through NFkB. It was reasonably argued that monocytes and myeloid dendritic cells were likely involved.

I think this is a valuable contribution because of its unique high quality data set, particularly if the data are accessible for open access mining. The analyses are sophisticated and make good points. I do have some small reservations as indicated above, particularly related to the rather small differences in antibody responses between the groups and the considerable overlap between individual antibody responses to vaccination. It might have been useful to compare the highest with the lowest fold-antibody response percentiles in these cohorts. There might be good reasons for not doing that but it might sharpen the transcriptional differences seen here or indeed reveal new ones.

Finally, are there any translational implications in this study? Should we be adding adjuvants to all vaccines to enhance antibody responses or just to some of them? Is there any epidemiological situation where this kind of transcriptional analysis could identify those who need a vaccine most, a higher dose or a different type of vaccine?

Author Rebuttal to Initial comments

Reviewers' Comments:

Reviewer #1:

1. The population is divided in clusters with low, medium and high level of pre-vaccination inflammatory transcripts. Can this be confirmed with evidence that pre-vaccination levels of pro-inflammatory cytokines in humans correlate with higher vaccine antibody response? Were cofounding effects such as use of corticosteroids considered in the study?

Reviewer #1 raises two good points. Concerning looking at cytokines as pre-vaccination correlates of antibody response, only a few of the studies included in this meta-analysis also generated plasma cytokines measurements. Below is an example of two of those studies where the endotypes are associated with different cytokines profiles pre-vaccination (Kruskal-Wallis test: $p \leq 0.05$). Due to the low absolute concentration of cytokines typically observed pre-vaccination, we do not think that these data are necessary to confirm the pre-vaccination endotypes that we have defined. Combined with the small number of studies with cytokine data available, we decided to not include this data in the manuscript.

Figure Annex 1. Soluble proteins with statistically significant differential expression between the endotypes pre-vaccination are shown for two studies part of the transcriptomic meta-analysis (top) Heatmap of the expression of the 5 soluble proteins distinguishing the three endotypes in study SDY1328 (hepatitis B vaccine study). The color gradient depicts the scaled protein concentration (mean of zero, a standard deviation of 1). The antibody response groups (top 30% and bottom 30% based on the D28 to D0 maximum fold-change) are shown as annotation of the heatmap. (bottom) Similar heatmap for SDY984 (Zoster vaccine study).

Concerning the comorbidities and corticosteroids usage, the participants included in this meta-analysis are healthy adults. Unfortunately, co-morbidities or the use of steroids is not part of the meta-data we have available on those participants (mention now at lines 462-464). We are planning a separate prospective study on a cohort where more extensive comorbidities and medication lists will be available which would allow us to address this comment.

2. In Figure 3, a reduced expression of pro-inflammatory genes after vaccination is observed for inflam.hi group as compared to inflam.low (including lower B cell response). What is the proposed mechanism that correlate the initial reduced immune stimulation with increase antibody response?

Reviewer #1 is correct in pointing out that inflam.hi participants have a less pronounced induction of inflammatory pathways in the first 3 days after vaccination compared to inflam.lo participants, which coincides with a lower magnitude of B cell response at day 7. One of the mechanisms we postulate that can explain why inflam.hi participants that have a lower B cell response in blood mount a higher antibody response at day 28 is the migration of antibody-secreting cells from the circulation to the tissue. We don't have tissue data for those participants to confirm this hypothesis. We added this point to the discussion lines 421-423.

3. Pre-existing inflammation in the elderly is characterized by reduced vaccine immune response. The same authors reported this effect in a population mostly 65-75 resulted in lower HBV vaccine response (<https://doi.org/10.1038/ncomms10369>). How the evidence reported in this manuscript can inform vaccine design for this population?

Reviewer #1 makes a good point about pre-vaccination inflammation being associated with hyporesponse to some vaccines in the elderly. The endotypes defined in this current study were not associated with antibody response to vaccination in the elderly. We explain the difference by showing that the inflammatory signature identified in this study has no overlapping gene with the one identified in the elderly, and thus likely reflects a distinct aspect of inflammation. This is discussed at lines 457-466 of the discussion.

4. The authors propose the use of adjuvants to stimulate NFkB pathway before vaccination to validate the proposed immunological mechanism. Are there evidences supporting this claim?

Reviewer #1 is bringing an important point. We unfortunately don't have experimental validation of the benefit of NFkB modulation pre-vaccination on vaccination but are actively generating experimental data (in organoid and mice models) that will be the subject of an independent manuscript. The results presented in this manuscript are supported by multiple independent cohorts despite the absence of additional experimental validation.

Reviewer #2:

1. A tremendous amount of work has gone in to the generation and analysis of the transcriptomic data- but immune data is not included. Specifically, the magnitude of the antibody or T cell response are missing – to substantiate that the magnitude of the transcriptomic signature is a close surrogate of the adaptive immune response.

We are thankful for reviewer #2's comment and apologize for the confusion. We have already included data from immunological assays. Results for the primary endpoint used in this study (i.e., vaccine specific antibody titers at day 28) are presented in Figure 4A as Maximum fold-change (D28/D0). Antibody response at day 28 post-vaccination was chosen because it was the only common endpoint shared across the vaccine studies included in this meta-analysis. Unfortunately, vaccine specific T cell responses were measured on a small number of participants (less 20) and thus are not presented in this manuscript. We have revised the axis title of Figure 4A to clarify this point.

2. Pre-vacc endotype differences explained 12.5% of variation in gene expression pre and post infection- approximating the level of variance captured by demographics alone. It would be helpful to show some supplemental data that the variance captured by both these sets of parameters are independent.

Reviewer #2 is bringing up a good point. Below is the Principal Variant Component Analysis of the demographics alone and the endotypes alone. Only a small difference in the variance explained is observed with the figure presented in the paper which suggests that the variance captured by both sets of parameters is indeed independent. This is now mentioned in the Results section at lines 249-250.

Figure Annex 3. PVCA with or without the endotypes. (left) Figure S3A (middle) PVCA with only the demographic variables (right) PVCA with only the endotypes.

3. The B cell signatures in Figure 3 are described as “dampened”. While this is clearly one interpretation- could they simply not be in the blood if there is an ongoing or recent infection? Perhaps “diminished in the periphery” could be used interchangeably? Some discussion would be helpful on this subject.

We agree with the reviewer and have edited the results (changed damped by lower at line 262) and discussion accordingly (lines 421-423).

4. The T cell data in Figure 3 supplemental seems quite exciting and should be considered for the main body.

We agree with reviewer #2 and have added this panel to Figure 3.

5. The relationship between the endotypes and antibody results at day 28 is buried in the supplemental data – perhaps as it did not reach statistical significance – but represents one of the only analyses that links directly to antibodies. It would be helpful to include this in the main body and any other analyses that directly relate the immunological endpoint of greatest interest in vaccine development (antibody titers).

We want to thank reviewer #2 for this comment. The primary endpoint used in this study is antibody response at day 28 which is presented in Figure 4. The y-axis legend of panel A was edited to clarify that it is an antibody response.

6. The differences in endotype prediction across vaccines is fascinating-particular in the context of particular vaccines. Were vaccines grouped into vectors, polysaccharides, those to which pre-existing immunity exists, etc. to enhance power and explore endotype predictive accuracy on response profiles? Some discussion/explanation would be very helpful to understand what is common about the vaccines for which the endotypes are highly predictive???. Are they more IFN dependent? Are they novel responses?

This comment by reviewer #2 is well justified. Figure 6a present the vaccines response tested for their associations with NFkB signaling (the main transcription factors demarcating the endotypes). The three vaccines for which the signatures did not predict response as accurately as the others (Tb, Pneumococcus, Smallpox) do not share a common vector, type of vaccine or have higher antibody titers pre-vaccination. The only commonality we could identify is the small number of samples for those vaccines. These points are mentioned in the Results (lines 301-309). We also added a supplementary table (Supplementary Table 1) with the list of vaccines included in the meta-analysis.

7. Figure 6a shows the program in the “high” endotype – which is striking. It would be helpful to see the other endotypes as well in supplemental- to visualize the universality of these profiles- that presumably would be present irrespective of the level of viral/bacterial exposure.

We agree with reviewer #2 and present in the boxplot in Figure 6a the viral/bacterial score in all three endotypes.

8. It is unclear whether the viral vs bacterial signatures relate to pathogens or commensals? Are the LPS/zymosan signatures related to pathogenic bacteria or simply to microbial translocation and/or microbiome shifts? Where signatures of microbial translocation also explored and does this explain this bacterial signature? This would be critical to explore/add. Presumably, cytokine/serum analytes were also captured in the HIPC and could add tremendous value to understanding this baseline advantage in the high endotype population.

Reviewer #2 makes a good point. Most bacteria used to develop the bacterial/viral signature described in Sweeney TE *et al.* are pathogenic. We edited the text to convey that information (lines 351). Unfortunately, cytokine profiling (see answer to reviewer #1) was only available for a small fraction of the participants preventing us from including them in this manuscript.

9. While antibodies represent the surrogate of protection against most infections, some discussion on whether these are mechanistic correlates and whether these mark durability differences should be mentioned. Reviewer #2 makes an important comment. We included as a supplementary figure antibody response at day 180 post-vaccination stratified by endotypes (Figure S4B). A similar trend was observed as on day 28 without reaching statistical significance. We believe it is due to the low sample size. Because we don't have enough data, we refrained from making a conclusive assessment of the association between the endotypes and antibody response durability in the paper.

Reviewer #4:

The scale on the y axis is not clear in Fig 4A, I assume it is on a log2 scale. I am not expert in interpreting ROC plots but the prediction doesn't look particularly strong. However when broken down for the inactivated influenza vaccine the p value was very impressive. The heterogeneity of some of the cohorts, the different types of vaccine and small numbers in some parts of the study may have lessened the overall effect. The strong proinflammatory response was similar to that stimulated by TLR ligands/adjuvants and is connected through NFkB. It was reasonably argued that monocytes and

myeloid dendritic cells were likely involved.

We agree with reviewer #4 and have edited the axis labels in Figure 4A as suggested.

I do have some small reservations as indicated above, particularly related to the rather small differences in antibody responses between the groups and the considerable overlap between individual antibody responses to vaccination. It might have been useful to compare the highest with the lowest fold-antibody response percentiles in these cohorts. There might be good reasons for not doing that but it might sharpen the transcriptional differences seen here or indeed reveal new ones.

We totally agree with reviewer #4 and for that reason we included only the top tercile and bottom tercile of antibody response when building our supervised classifier (lines 290-292).

Finally, are there any translational implications in this study? Should we be adding adjuvants to all vaccines to enhance antibody responses or just to some of them? Is there any epidemiological situation where this kind of transcriptional analysis could identify those who need a vaccine most, a higher dose or a different type of vaccine?

This comment by reviewer #4 is highly relevant. We amended the discussion and added a section describing how the endotypes can lead to translational implications for vaccination (lines 481-489).

Decision Letter, first revision:

Subject: Your manuscript, NI-A33975A

Message: Our ref: NI-A33975A

23rd Aug 2022

Dear Dr. Sekaly,

Thank you for your patience as we've prepared the guidelines for final submission of your Nature Immunology manuscript, "Pan-vaccine analysis reveals innate immune endotypes predictive of antibody responses to vaccination" (NI-A33975A). Please carefully follow the step-by-step instructions provided in the attached file, and add a response in each row of the table to indicate the changes that you have made. Please also check and comment on any additional marked-up edits we have proposed within the text. Ensuring that each

point is addressed will help to ensure that your revised manuscript can be swiftly handed over to our production team.

When you upload your final materials, please include a point-by-point response to any remaining reviewer comments and please make sure to upload your checklist.

If you have not done so already, please alert us to any related manuscripts from your group that are under consideration or in press at other journals, or are being written up for submission to other journals (see: <https://www.nature.com/nature-portfolio/editorial-policies/plagiarism#policy-on-duplicate-publication> for details).

In recognition of the time and expertise our reviewers provide to Nature Immunology's editorial process, we would like to formally acknowledge their contribution to the external peer review of your manuscript entitled "Pan-vaccine analysis reveals innate immune endotypes predictive of antibody responses to vaccination". For those reviewers who give their assent, we will be publishing their names alongside the published article.

Nature Immunology offers a Transparent Peer Review option for new original research manuscripts submitted after December 1st, 2019. As part of this initiative, we encourage our authors to support increased transparency into the peer review process by agreeing to have the reviewer comments, author rebuttal letters, and editorial decision letters published as a Supplementary item. When you submit your final files please clearly state in your cover letter whether or not you would like to participate in this initiative. Please note that failure to state your preference will result in delays in accepting your manuscript for publication.

Cover suggestions

As you prepare your final files we encourage you to consider whether you have any images or illustrations that may be appropriate for use on the cover of Nature Immunology.

Nature Immunology has now transitioned to a unified Rights Collection system which will allow our Author Services team to quickly and easily collect the rights and permissions required to publish your work. Approximately 10 days after your paper is formally accepted, you will receive an email in providing you with a link to complete the grant of rights. If your paper is eligible for Open Access, our Author Services team will also be in touch regarding any additional information that may be required to arrange payment for your article.

Please note that *Nature Immunology* is a Transformative Journal (TJ). Authors may publish their research with us through the traditional subscription access route or make their paper immediately open access through payment of an article-processing charge (APC). Authors will not be required to make a final decision about access to their article until it has been accepted. [Find out more about Transformative Journals](https://www.springernature.com/gp/open-research/transformative-journals).

If you have any questions about costs, Open Access requirements, or our legal forms, please contact ASJournals@springernature.com.

Please use the following link for uploading these materials: [REDACTED]

Best regards,

Elle Morris
Senior Editorial Assistant
Nature Immunology
Phone: 212 726 9207
Fax: 212 696 9752
E-mail: immunology@us.nature.com

On behalf of

Jamie D.K. Wilson, D.Phil
Chief Editor
Nature Immunology
212 726 9207
j.wilson@us.nature.com

Reviewer #2:

Remarks to the Author:

The authors have addressed most the reviewer's comments - including the referral to Figure 4 containing the primary endpoints - antibody responses. It is striking that the reader must wait until figure 4 to see the variability in responses - and it is not clear for each vaccine if binding titers or neutralizing titers were used in this study as a the endpoint and whether these are the same endpoints that are used to measure vaccine response/protection clinically. This should be addressed as a final point in the methods and results.

Author Rebuttal, first revision:

Reviewer #2 (Remarks to the Author):

The authors have addressed most the reviewer's comments - including the referral to Figure 4 containing the primary endpoints - antibody responses. It is striking that the reader must wait until figure 4 to see the variability in responses - and it is not clear for each vaccine if binding titers or neutralizing titers were used in this study as a the endpoint and whether these are the same endpoints that are used to measure vaccine response/protection clinically. This should be addressed as a final point in the methods and results.

We want to thank reviewer #2 for this comment. We provide in Supplementary Table 1 the list of assays used to assess antibody response for each study. Below are panels A and C stratified by assays. The associations between the inflammatory endotypes and antibody response or the classifier and antibody response groups were not statistically significantly different between assays used to measure antibody response. To clarify this point, we now mention in the result section that the list of assays is provided in Supplementary Table 1 (lines 243-244). We added a sentence mentioning that the endotypes were not associated with the antibody response assays at lines 254-255. We added that the classifier was not associated with the antibody response assays at lines 273-274.

Figure 4A. Boxplot of the maximum fold-change (MFC) antibody responses as a function of the pre-vaccination inflammation endotypes stratified by antibody response assay. The MFC was scaled to a mean of 0 and a standard deviation of 1 across vaccines. A Kruskal-Wallis test was used to assess differences in antibody response between endotypes and resulted in a p-value of 0.118, 0.211, and 0.0463, for ELISA, HAI, and neutralizing assays, respectively.

Figure 4C. The top 500 predictive genes/features included in the classifier (importance > 0%) overlapped with inflammatory genes identified in the unsupervised approach (Fisher's exact test: $p=8.98 \times 10^{-10}$). Heatmap showing the pre-vaccination expression of the overlapping genes. Samples (columns) are ordered by increasing expression level of the inflammatory genes. A Wilcoxon-rank sum test was used to assess the association between the inflammatory signatures and high/low antibody response and resulted in a p-value of 0.0412, 0.0993, and 0.0314 for ELISA, HAI, and neutralizing assays, respectively.

Final Decision Letter:

In reply please quote: NI-RS33975B

Dear Dr. Sekaly,

I am delighted to accept your manuscript entitled "Pan-vaccine analysis reveals innate immune endotypes predictive of antibody responses to vaccination" for publication in an upcoming issue of *Nature Immunology*.

Over the next few weeks, your paper will be copyedited to ensure that it conforms to *Nature Immunology* style. Once your paper is typeset, you will receive an email with a link to choose the appropriate publishing options for your paper and our Author Services team will be in touch regarding any additional information that may be required.

Please note that *Nature Immunology* is a Transformative Journal (TJ). Authors may publish their research with us through the traditional subscription access route or make their paper immediately open access through payment of an article-processing charge (APC). Authors will not be required to make a final decision about access to their article until it has been accepted. Find out more about Transformative Journals.

Authors may need to take specific actions to achieve compliance with funder and institutional open access mandates. If your research is supported by a funder that requires immediate open access (e.g.

according to Plan S principles) then you should select the gold OA route, and we will direct you to the compliant route where possible. For authors selecting the subscription publication route, the journal's standard licensing terms will need to be accepted, including self-archiving policies. Those licensing terms will supersede any other terms that the author or any third party may assert apply to any version of the manuscript.

Your paper will be published online soon after we receive your corrections and will appear in print in the next available issue. Content is published online weekly on Mondays and Thursdays, and the embargo is set at 16:00 London time (GMT)/11:00 am US Eastern time (EST) on the day of publication. Now is the time to inform your Public Relations or Press Office about your paper, as they might be interested in promoting its publication. This will allow them time to prepare an accurate and satisfactory press release. Include your manuscript tracking number (NI-RS33975B) and the name of the journal, which they will need when they contact our office.

About one week before your paper is published online, we shall be distributing a press release to news organizations worldwide, which may very well include details of your work. We are happy for your institution or funding agency to prepare its own press release, but it must mention the embargo date and Nature Immunology. Our Press Office will contact you closer to the time of publication, but if you or your Press Office have any enquiries in the meantime, please contact press@nature.com.

Also, if you have any spectacular or outstanding figures or graphics associated with your manuscript - though not necessarily included with your submission - we'd be delighted to consider them as candidates for our cover. Simply send an electronic version (accompanied by a hard copy) to us with a possible cover caption enclosed.

If you have not already done so, we strongly recommend that you upload the step-by-step protocols used in this manuscript to the Protocol Exchange. Protocol Exchange is an open online resource that allows researchers to share their detailed experimental know-how. All uploaded protocols are made freely available, assigned DOIs for ease of citation and fully searchable through nature.com. Protocols can be linked to any publications in which they are used and will be linked to from your article. You can also establish a dedicated page to collect all your lab Protocols. By uploading your Protocols to Protocol Exchange, you are enabling researchers to more readily reproduce or adapt the methodology you use, as well as increasing the visibility of your protocols and papers. Upload your Protocols at www.nature.com/protocolexchange/. Further information can be found at www.nature.com/protocolexchange/about .

Please note that we encourage the authors to self-archive their manuscript (the accepted version before copy editing) in their institutional repository, and in their funders' archives, six months after publication. Nature Portfolio recognizes the efforts of funding bodies to increase access of the research they fund, and strongly encourages authors to participate in such efforts. For information about our editorial policy, including license agreement and author copyright, please visit www.nature.com/ni/about/ed_policies/index.html

Sincerely,

Jamie D.K. Wilson, D.Phil
Chief Editor
Nature Immunology
212 726 9207
j.wilson@us.nature.com